# META-KNOWLEDGE EXTRACTION: UNCERTAINTY-AWARE PROMPTED META-LEARNING

## ABSTRACT

Conventional meta-learning typically involves adapting all meta-knowledge to specific tasks, which incurs high computational costs due to the adaption process. To address this limitation, we introduce a more efficient gradient-based meta-learning framework called Uncertainty-Aware Prompted Meta-Learning (UAPML). Instead of adapting the entire meta-knowledge, we introduce a meta-knowledge extraction paradigm inspired by the success of large language models. In this paradigm, we freeze the model backbone and employ task-specific prompts to extract meta-knowledge for few-shot tasks. To construct the task-specific prompts, a learnable Bayesian meta-prompt is employed to provide an ideal initialization. Through theoretical analysis, we demonstrate that the posterior uncertainty of the Bayesian meta-prompt aligns with that of the task-specific prompt, which can be used to modulate the construction of task-specific prompts. Accordingly, we propose two ways, i.e., the soft and hard way, to automatically construct task-specific prompts from the meta-prompt when dealing with new tasks. Experimental results demonstrate the efficiency of the meta-knowledge extraction paradigm and highlight the significantly reduced computational cost achieved by our UAPML framework without the degradation of performance.

## 1 INTRODUCTION

Meta-learning (Hospedales et al., 2021; Huisman et al., 2021) has gained significant attention for its practicality in scenarios when only a limited number of labeled data are available. This techique inductively transfers meta-knowledge, such as parameter initialization, among different tasks, thereby improving data efficiency in low-resource scenarios. Most successful methods (Finn et al., 2017; Lee et al., 2019; Rusu et al., 2019), particularly optimization-based methods, mainly focus on adapting the meta-knowledge to task-specific knowledge, leading to different model variants for each specific task. Although such a knowledge adaption paradigm has yielded promising results, it comes at a high computational cost during the adaption process (Nichol et al., 2018).

The recent surge in popularity of large-scale pre-trained models such as ChatGPT and GPT-4 show a more efficient way to deal with tasks through knowledge extraction (Brown et al., 2020; Liu et al., 2023a). In contrast to knowledge adaption, knowledge extraction employs prompts (e.g., continuous tensors) that are prepended to the input, to bridge the gap between the general knowledge and task-specific knowledge. This efficient paradigm freezes the model backbone, eliminating the need for separate copies of the model for each task and significantly reducing the computational cost required for adaption on various tasks (Lester et al., 2021).

**Therefore, we seek to leverage the knowledge extraction to overcome the computational inefficiency of the meta-knowledge adaption process (see Fig. 1).** Applying knowledge extraction in the context of meta-learning requires two points (Liu et al., 2023b): (1) the compatibility of captured meta-knowledge across different tasks during the meta-training phase; (2) an appropriate prompt that effectively captures task-specific differences to enable task-specific knowledge extraction.

In response, we propose a novel **Uncertainty-Aware Prompted Meta-learning** (abbreviated as UAPML), which employs prompts for bridging the gap between general meta-knowledge and task-specific knowledge. In our UAPML framework, we share the entire model backbone (i.e., the feature extractor) among tasks to guarantee the compatibility of captured meta-knowledge, and design the task-specific prompt to help capture task-specific information. Sharing the backbone is supported by

Figure 1: The comparison between meta-knowledge adaption and meta-knowledge extraction.

the previous findings that highlight the potential of shared features in the success of meta-learning (Raghu et al., 2019). It is also recognized that a fixed feature representation may not adequately cater to the diverse requirements of different tasks (Oh et al., 2020; Arnold & Sha, 2021), which confirms the need for the task-specific prompt. These task-specific prompts are continuous tensors that can be learned from available data, providing task-specific extraction in addition to the shared feature. However, learning distinct prompts from a large number of few-shot tasks is impractical and inefficient, and over-fitting easily occurs under the few-shot scenario. This pushes us to propose a Bayesian meta-prompt that provides an ideal initialization for task-specific prompts. Such a Bayesian treatment for meta-prompt also offers a crucial advantage of an uncertainty measure to reflect the relationship among tasks. Specifically, the posterior uncertainty of each dimension in the meta-prompt aligns with the uncertainty of task-specific prompts. With theoretical analysis, we show that highly shared dimensions among tasks exhibit low variance, indicating low uncertainty, while dimensions that are diverse and specific to certain tasks display higher variance, indicating higher uncertainty. Building upon this analysis, we introduce two automatic ways to derive the task-specific prompts from the meta-prompt: soft and hard modulation. The soft modulation adjusts the learning rate for each dimension when handling specific tasks, while the hard modulation freezes parts of dimensions for sharing and selectively tunes the remaining dimensions for specific tasks.

Our contributions can be summarized as follows. (1) We propose the Uncertainty-Aware Prompt Meta-learning (UAPML) framework, which improves the efficiency of gradient-based meta-learning by focusing on meta-knowledge extraction for few-shot tasks instead of computationally expensive meta-knowledge adaption (2) We provide a theoretical understanding of the alignment between Bayesian meta-prompt and task-specific prompt, enabling the construction of task-specific prompts that balance shared and task-specific information, enhancing flexibility in handling few-shot tasks (3) Based on our theoretical findings, we introduce two innovative methods for designing task-specific prompts using the Bayesian treatment of the meta-prompt. The soft and hard modulation techniques automatically generate task-specific prompts while considering the shared and task-specific information among the few-shot tasks. (4) We conduct extensive experiments in various settings, demonstrating that UAPML achieve a comparable or even prior performance over several state-of-the-art meta-learning methods when significantly reducing the computation consumption.

## 2 RELATED WORK

**Meta-Learning** Meta-learning encompasses three main categories: optimization-based (Finn et al., 2017), model-based (Ha et al., 2016) and metric-based (Snell et al., 2017). We focus on the well-performed optimization-based meta-learning. These methods employ a meta-knowledge adaptation paradigm where all meta-knowledge is quickly adapted to specific tasks through gradient descent. However, this paradigm can be computationally expensive as all meta-knowledge needs to be adapted for each task. Efforts have been made to mitigate this burden. One (Nichol et al., 2018) uses first-order approximation, while another (Li et al., 2017) acquires an appropriate learning rate for one-step update adaptation. Recent work Raghu et al. (2019) has highlighted the significance of feature reuse in meta-learning success. By freezing most parameters except the prediction head, computational consumption can be significantly reduced. Another study (Von Oswald et al., 2021) follows a similar idea but incorporates sparse adaptation with additional masking parameters, enabling specific task adaptation of parameter subsets. While frozen representations have been effective in some cases, recent research by Oh et al. (2020) suggests that they may not adequately handle the complexity of real-world scenarios, highlighting the importance of embedding adaptation (Arnold & Sha, 2021). This trade-off between effectiveness and efficiency necessitates a solution. In our work, we address this challenge by incorporating prompts into the meta-learning framework. This approach provides an efficient means to capture task-specific information, bridging the gap between reused embeddings and the requirements of individual tasks.

Some other efforts also focus on capturing task-specific information, but their aims do not lie in efficiency. These efforts can be divided into two categories: mixture meta-knowledge (Yao et al., 2019; Jerfel et al., 2019; Yao et al., 2020; Zhang et al., 2021b; Wu et al., 2023) and conditional meta-learning (Rusu et al., 2019; Wang et al., 2020; Denevi et al., 2020). The mixture meta-knowledge approach associates clusters of similar tasks with different components, resulting in increased consumption proportional to the number of clusters. On the other hand, conditional meta-learning methods maintain a single meta-knowledge but need to design the additional architecture to customize the transfer of the meta-knowledge according to target tasks. For example, Rusu et al. (2019) proposes to employ an encoder-decoder architecture to construct task-specific classification heads, while Oreshkin et al. (2018) employs task-dependent scaled metric to customize the task-specific metric space. More recently, Baik et al. (2020) and Liu et al. (2020) employed a generative network to provide customized hyperparameters, such as the learning rate and the regularization weight, conditioned on the training state of the task. Since their aims do not lie in efficiency, their high consumption might be a concern. Instead, our proposed UAPML takes into account both effectiveness and efficiency, allowing a comparable or even better performance with less consumption.

Bayesian meta-learning is another relevant area. Previous efforts (Yoon et al., 2018; Gordon et al., 2019; Ravi & Beatson, 2019) have demonstrated the effectiveness of Bayesian treatment. Similarly, we employ Bayesian treatment on our meta-prompt but further conduct a comprehensive analysis of the benefits of the introduced uncertainty measure. Building upon the analysis, we propose two ways to construct the task-specific prompts, benefiting from the additional uncertainty measure.

**Prompt Tuning**  Knowledge extraction (i.e., knowledge probing) (Petroni et al., 2019; Davison et al., 2019) is an important work in the language models. It involves using language triggers to induce the model to generate relational facts. Prompts, one important tool in knowledge extraction, is introduced in GPT-3 (Brown et al., 2020) and showed that can be designed or learned to capture the task-related information (Liu et al., 2021a). Schick & Schütze (2021) introduce pattern-exploiting training, demonstrating that providing task descriptions to pre-trained models can achieve comparable performance to standard supervised fine-tuning. Beyond discrete vocabulary spaces, many works try to learn prompt vectors in continuous spaces (Liu et al., 2021b; Li & Liang, 2021; Zhang et al., 2021a). These techniques, collectively referred to as prompt tuning (Lester et al., 2021; Gu et al., 2022), aim to optimize prompt representations for specific tasks, such as the semantic prompt captured from the class name Chen et al. (2023). In this paper, we are inspired by knowledge extraction and try to propose a more efficient meta-learning method, but not requiring extra context.

Efforts to integrate meta-learning with prompt tuning have been made in NLP. Hou et al. (2022) employ meta-learning to learn a better initialization while Jiang et al. (2023) further propose to construct a prompt pool. Different from the existing prompt tuning in NLP, we consider the meta-prompt as a stochastic high-dimensional variable, serving as an initialization for task-specific prompts. Additionally, we theoretically demonstrate that the posterior uncertainty of the meta-prompt can effectively guide the construction of task-specific prompts, which is a more efficient approach compared to prompt pools that necessitate linearly increasing resources.

## 3 PRELIMINARY

**Model-Agnostic Meta-learning**  Model-agnostic meta-learning (MAML) (Finn et al., 2017) is one of the most popular meta-learning methods. It employs a bi-level optimization framework to enable fast adaption of models across various few-shot tasks. Specifically, MAML aims to learn a decent initialization of model parameters, i.e., meta-knowledge, through the outer loop, while fine-tuning the model parameters to specific tasks through the inner adaption. This paradigm of adjusting meta-knowledge for different tasks can be considered as a knowledge adaption paradigm. Mathematically, let $\boldsymbol{\theta}$ be the initialization of model parameters, the inner adaption process of MAML yields a task-specific parameter $\boldsymbol{\theta}'$ for a few-shot task $\tau$ through one or more gradient descent updates, where one gradient descent update can be denoted as: $\boldsymbol{\theta}' = \boldsymbol{\theta} - \alpha\nabla_{\boldsymbol{\theta}}\mathcal{L}_{\tau}(\boldsymbol{\theta})$, where $\alpha$ is the learning rate for the inner adaption and $\mathcal{L}_{\tau}$ is the task-specific loss function. The optimization objective of MAML is defined over a number of tasks: $\mathcal{L}(\boldsymbol{\theta}) = \int_{\tau\in p(\tau)}\mathcal{L}_{\tau}(\boldsymbol{\theta}') = \int_{\tau\in p(\tau)}\mathcal{L}_{\tau}(\boldsymbol{\theta} - \alpha\nabla_{\boldsymbol{\theta}}\mathcal{L}_{\tau}(\boldsymbol{\theta}))$. The objective is optimized via the outer loop: $\boldsymbol{\theta} = \boldsymbol{\theta} - \beta\nabla_{\boldsymbol{\theta}}\mathcal{L}(\boldsymbol{\theta})$, where $\beta$ is the learning rate.

**Prompt Tuning**  Prompt tuning (Lester et al., 2021; Gu et al., 2022) has emerged as a promising method for knowledge extraction, which utilizes pre-trained language models for various down-

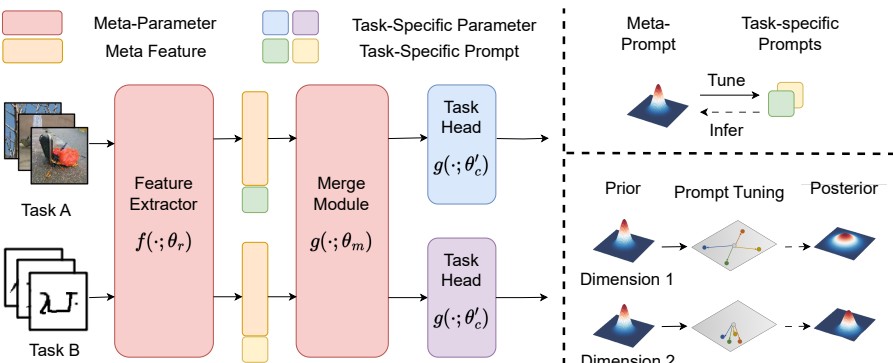

Figure 2: The architecture diagram of UAPML (**left**) and an intuitive illustration of how the uncertainty of the posterior meta-prompt is used to modulate the tuning of task-specific prompts (**right**).

stream tasks in the field of natural language processing. Instead of fine-tuning pre-trained language models on specific tasks, prompt tuning methods freeze the model backbone and incorporate task-specific prompts as additional inputs to effectively handle different tasks. The prompt is typically designed as a sequence of discrete tokens or implemented as a trainable continuous vector $s$, which is then concatenated to the input of a pre-trained model to provide task-specific guidance. Specifically, suppose a pre-trained model $\mathcal{M}$ conducts a mapping $\pi : \mathcal{X} \rightarrow \mathcal{H}$, where $\mathcal{X}$ is the input space and $\mathcal{H}$ is the representation space. Normally, the pre-trained model outputs representations by: $h = \pi(X)$, while the prompted model outputs representations by $\tilde{h} = \pi(s||X)$, where $||$ denotes the concatenation of inputs and prompts. The advantage of prompt tuning is that we do not need to update the parameters of the pre-trained model $\mathcal{M}$ but just learn different prompts when dealing with various downstream tasks.

## 4 METHOD

In what follows, we introduce the details of the proposed Uncertainty-Aware Prompted Meta-Learning (UAPML), which is shown in Fig. 2.

### 4.1 PROMPTED META-LEARNING

Let the model parameters $\theta$ consist of two parts, the model backbone $\theta_r$ (e.g., the feature extractor) and the task classification head $\theta_c$. Following the existing work (Raghu et al., 2019), our UAPML freezes the model backbone $\theta_r$ and only adapts the classification head $\theta_c$ to the specific tasks as $\theta'_c$. This approach is based on the assumption, consistent with (Raghu et al., 2019), that the features extracted from the model backbone possess general meta-knowledge across various downstream few-shot tasks and should be reused. To facilitate the feature reuse considering the task-specific information, we employ a task-specific prompt to guide how to reuse the feature (i.e., the meta-knowledge extraction process) for a specific task.

#### 4.1.1 META-PROMPT

Since meta-learning models are designed to tackle a multitude of few-shot tasks, it would be inefficient or impractical to learn distinct prompts for each task. Additionally, directly learning such prompts on limited data can lead to overfitting issues. Therefore, following the bi-level optimization paradigm of meta-learning, we introduce a learnable meta-prompt $s \in \mathbb{R}^n$, which serves as a decent initialization for task-specific prompts. To better adapt to real-world scenarios where tasks can significantly differ from each other, we employ a Bayesian treatment for the meta-prompt. We model the meta-prompt as a random variable instead of a deterministic vector. This Bayesian treatment allows us to capture the uncertainty associated with the meta-prompt and provides a flexible initialization for task-specific prompts. Such uncertainty measure introduced by Bayesian treatment can avoid conflict among tasks and help the model to better handle the variations in few-shot tasks.

Specifically, given a globally-shared meta-prompt distribution $p(s)$, we sample a meta-prompt from the distribution to provide initialization for task-specific prompts. For a specific downstream task,

the meta-prompt is tuned with one or more steps of gradient descent updates:

$$\boldsymbol{s}' = \boldsymbol{s} - \alpha \nabla_{\boldsymbol{s}} \mathcal{L}_{\tau}(\boldsymbol{s}, \boldsymbol{\theta}), \quad \boldsymbol{s} \sim p(\boldsymbol{s}). \tag{1}$$

To incorporate task-specific guidance, the task-specific prompts $\boldsymbol{s}'$ are concatenated with the features from the backbone model $\boldsymbol{\theta_r}$. This concatenation combines the knowledge encoded in the prompts and the features, enabling the model to leverage both sources of information for each specific task. Moreover, to fuse the information from the prompt and the backbone model, we introduce a merge module. The merge module is parameterized by a neural network with parameters $\boldsymbol{\theta}_m$. It takes both the task-specific prompts and the features as inputs. Such a merge module is designed to capture the complementary information between the prompts and the features, allowing them to effectively work together to improve the model's performance on the specific task:

$$\hat{\boldsymbol{y}} = g(f(\boldsymbol{x}; \boldsymbol{\theta}_r) || \boldsymbol{s}'; \boldsymbol{\theta}_m, \boldsymbol{\theta}'_c), \tag{2}$$

where $f$ is the feature extractor and $g$ is the merged layer with a task-specific classification head $\boldsymbol{\theta}'_c$. And the task-specific classification head is attained following:

$$\boldsymbol{\theta}'_c = \boldsymbol{\theta}_c - \alpha \nabla_{\boldsymbol{\theta}_c} \mathcal{L}_{\tau}(\boldsymbol{s}, \boldsymbol{\theta}), \quad \boldsymbol{s} \sim p(\boldsymbol{s}). \tag{3}$$

Note that UAPML freezes the model backbone $\boldsymbol{\theta}_r$, occupying most of the parameters, as well as a small merge layers $\boldsymbol{\theta}_m$. **Only the meta-prompt $s$ and the classification head $\theta_c$ are updated for a specific task.** In this way, the computational consumption is significantly reduced. The analysis of the time complexity is shown in Appendix A.3. By combining the task-specific prompts, the features from the backbone model, and the merge module, the model can leverage the prompt-guided knowledge from the rich representations captured by the backbone model to make accurate predictions on the task at hand.

### 4.1.2 INFERENCE ALGORITHM

With a set of meta-learning tasks $\mathcal{D}$, the Bayesian treatment on meta-prompt brings the issue of the intractability of the posterior $p(\boldsymbol{s}|\mathcal{D})$. To address this issue, we employ variational inference to approximately infer the posterior. Specifically, we approximate the posterior $p(\boldsymbol{s}|\mathcal{D})$ with a variational distribution $q(\boldsymbol{s}; \boldsymbol{\phi})$. The evidence lower bound is defined as:

$$\log p(\mathcal{D}; \boldsymbol{\theta}) \geq \mathbb{E}_{q(\boldsymbol{s}; \boldsymbol{\phi})} \left[ \log p(\mathcal{D}|\boldsymbol{s}; \boldsymbol{\theta}) \right] - \mathbf{KL} \left[ q(\boldsymbol{s}; \boldsymbol{\phi}) || p(\boldsymbol{s}) \right], \tag{4}$$

where $KL(\cdot||\cdot)$ is the Kullback–Leibler divergence (KL-divergence) and $\log p(\mathcal{D}|\boldsymbol{s}; \boldsymbol{\theta})$ is the log-likelihood of data conditioned on the meta-prompt:

$$p(\mathcal{D}|\boldsymbol{s}; \boldsymbol{\theta}) = \int_{\tau \in p(\tau)} -\mathcal{L}_{\tau}(\boldsymbol{s}', \boldsymbol{\theta}), \tag{5}$$

where $\boldsymbol{s}'$ is the task-specific prompt after a few steps of gradient descent from the meta-prompt $\boldsymbol{s} \sim q(\boldsymbol{s}; \boldsymbol{\phi})$. Note that we assume $q(\boldsymbol{s}; \boldsymbol{\phi})$ to be a Gaussian with diagonal covariance, which is parameterized by $\boldsymbol{\phi} = \boldsymbol{\mu}, \boldsymbol{\rho}$. The standard deviation is parameterized as $\boldsymbol{\sigma} = \log(1 + \exp(\boldsymbol{\rho}))$ to avoid the invalid value (i.e., the negative value). The detailed derivation is shown in Appendix A.1.

### 4.2 THEORETICAL ANALYSIS

The Bayesian meta-prompt serves as a valuable initialization for efficiently constructing task-specific prompts. However, the SGD-based construction approach treats each dimension equally, which may not be optimal (Kingma & Ba, 2014). This is because different dimensions of the meta-prompt can contain different amounts of shared information across tasks and require different levels of tuning to capture task-specific information. In this subsection, we present a theoretical analysis that demonstrates the alignment between the uncertainty of the meta-prompt and task-specific prompt when optimizing the objective Eq. 4. This finding motivates us to leverage the uncertainty of the meta-prompt to modulate the construction of task-specific prompts. In what follows, we provide the mathematical definition and the proof of this finding. More details are in Appendix A.2.

**Definition 1** *Let $H(\cdot)$ be the entropy, which measures the uncertainty of the random variable. $H(\boldsymbol{s})$ and $H(\boldsymbol{s}'|\boldsymbol{s}, \mathcal{D})$ describe the uncertainty of meta-prompt and the task-specific prompts respectively.*

**Proposition 1** *The uncertainty of the posterior of meta-prompt $H(\boldsymbol{s}_d)$ is in alignment with the uncertainty of task-specific prompts among tasks $H(\boldsymbol{s}'_d|\boldsymbol{s}_d, \mathcal{D})$ on each dimension d.*

To better present our analysis, we use another view of the optimization objective but equal to Eq. 4 following (Esmaeili et al., 2019). The objective of our proposed UAPML is to minimize the KL-divergence of each variable between the approximate posterior and the prior, which equals maximizing the negative KL-divergence (the lower bound $\mathcal{L}_{LB}$) and then can be derived as follows:

$$\mathcal{L}_{LB}(\boldsymbol{\theta}, \boldsymbol{\phi}) = -\mathbf{KL}\left[q(\mathcal{D}, \boldsymbol{s}', \boldsymbol{s}; \boldsymbol{\theta}, \boldsymbol{\phi}) \| p(\mathcal{D}, \boldsymbol{s}', \boldsymbol{s})\right] = \mathbb{E}_{q(\mathcal{D}, \boldsymbol{s}', \boldsymbol{s}; \boldsymbol{\theta}, \boldsymbol{\phi})}\left[\log \frac{p(\mathcal{D}|\boldsymbol{s}', \boldsymbol{s})}{p(\mathcal{D})} - \log \frac{q(\boldsymbol{s}', \boldsymbol{s}|\mathcal{D}; \boldsymbol{\theta}, \boldsymbol{\phi})}{q(\boldsymbol{s}', \boldsymbol{s}; \boldsymbol{\theta}, \boldsymbol{\phi})}\right]$$
$$- \mathbf{KL}\left[q(\mathcal{D})\|p(\mathcal{D})\right] - \mathbf{KL}\left[q(\boldsymbol{s}'|\boldsymbol{s}; \boldsymbol{\theta})\|p(\boldsymbol{s}'|\boldsymbol{s})\right] - \mathbf{KL}\left[q(\boldsymbol{s}; \boldsymbol{\phi})\|p(\boldsymbol{s})\right], \tag{6}$$

where $q(D)$ and $p(D)$ are the accessible and the inaccessible data distribution in the real world, respectively (Esmaeili et al., 2019). Note that the expectation of the term $\log \frac{q(\boldsymbol{s}', \boldsymbol{s}|\mathcal{D}; \boldsymbol{\theta}, \boldsymbol{\phi})}{q(\boldsymbol{s}', \boldsymbol{s}; \boldsymbol{\theta}, \boldsymbol{\phi})}$ in Eq. 6 equals to the mutual information $I(\boldsymbol{s}', \boldsymbol{s}; \mathcal{D})$ between prompts conditioned on the dataset $\mathcal{D}$ and not, and is derived as:

$$I(\boldsymbol{s}', \boldsymbol{s}; \mathcal{D}) = H(q(\boldsymbol{s}'|\boldsymbol{s}, \mathcal{D}; \boldsymbol{\theta})) - H(q(\boldsymbol{s}'|\boldsymbol{s}; \boldsymbol{\theta})) \approx H(q(\boldsymbol{s}'|\boldsymbol{s}, \mathcal{D}; \boldsymbol{\theta})) - H(q(\boldsymbol{s}; \boldsymbol{\phi})) \tag{7}$$

Note that the term $q(\boldsymbol{s}'|\boldsymbol{s}, \mathcal{D}; \boldsymbol{\theta})$ denotes the task-specific prompts given the meta-prompt among a set of meta-train tasks, as defined in Eq. 1. Since the term $q(\boldsymbol{s}'|\boldsymbol{s}; \boldsymbol{\theta})$ can be seen as the ones without tuning, it equals to the meta-prompt that is utilized as the initialization, i.e., $q(\boldsymbol{s}'|\boldsymbol{s}; \boldsymbol{\theta}) \approx q(\boldsymbol{s}; \boldsymbol{\phi})$. This approximation supports the derivation of Eq. 7. Furthermore, the derived result can be decomposed into each dimension based on our independence dimension assumption:

$$H(q(\boldsymbol{s}'|\boldsymbol{s}, \mathcal{D}; \boldsymbol{\theta})) - H(q(\boldsymbol{s}; \boldsymbol{\theta})) = \sum_d H(q(\boldsymbol{s}'_d|\boldsymbol{s}_d, \mathcal{D}; \boldsymbol{\theta})) - H(q(\boldsymbol{s}_d; \boldsymbol{\theta})) \tag{8}$$

When a specific dimension of the task-specific prompts exhibits diversity among a set of tasks, the corresponding dimension of the meta-prompt will have a larger uncertainty, which is mainly reflected in a larger standard deviation in our Gaussian distribution assumption. This alignment of uncertainty provides a way to effectively modulate the tuning of task-specific prompts, to allow the dimension with larger uncertainty to capture more task information while allowing the dimensions with smaller uncertainty to maintain the common information.

## 4.3 UNCERTAINTY-AWARE CONSTRUCTION FOR PROMPTS

Based on the aforementioned analysis, we propose two approaches, namely the soft and hard modulation, to effectively construct task-specific prompts from the meta-prompt. Note that these prompt construction methods are specifically designed for meta-test tasks, where the posterior distribution of the meta-prompt has already been inferred from a number of tasks.

**Soft Modulation** The soft modulation aims to encourage dimensions in the meta-prompt that contain more common information (i.e., dimensions with low uncertainty) to remain unchanged, thereby enabling information sharing across tasks. On the other hand, dimensions that share less common information are allowed to tune more, enabling the capture of task-specific information. Specifically, the soft construction regulates the learning rate on each dimension when tuning the prompts to specific tasks. We use $1/\boldsymbol{\sigma}_d$ to regulate the learning rate where $\boldsymbol{\sigma}_d$ is the standard deviation of $d$-th dimension of the meta-prompt:

$$h_{\text{soft}}(\alpha, \boldsymbol{\rho}) = \alpha/(1/\boldsymbol{\sigma}) = \alpha \log(1 + \exp(\boldsymbol{\rho})). \tag{9}$$

Such a method increases the learning rate of the dimensions with a large uncertainty to encourage to capture more task information while limiting the changes of the dimension with a low uncertainty to maintain the common information.

**Hard Modulation** The hard modulation approach follows a similar principle to the soft modulation but takes a more aggressive approach. It preserves dimensions in the meta-prompt with low uncertainty while allowing dimensions with higher uncertainty to freely adjust for task-specific information. It is similar to weight pruning (Han et al., 2015), a technique commonly used in network compression (Li et al., 2022), continual learning (Kang et al., 2022) and meta-learning (Von Oswald et al., 2021) to reduce computation and avoid over-parameterization. The previous weight tuning in meta-learning is achieved by freezing some weights in the inner adaption (Von Oswald et al., 2021) via learnable scores that weigh the importance of weights. This can lead to training instability due to the dynamic interaction between the scores and model parameters (Rosenbaum et al., 2019). Instead, we utilize the posterior inferred from meta-training, which is fixed when dealing with new tasks, to score the weights and make pruning decisions. Specifically, we employ the signal-to-noise

ratio (SNR) (Blundell et al., 2015) , calculated as $\frac{|\boldsymbol{\mu}|}{\sigma}$, to determine the amount of common information. The SNR takes into account both the standard deviation and mean value of the weights, where a larger mean indicates a more important role when merged with the representation. Consequently, weights with a large SNR will have a learning rate of zero to preserve the common information:

$$h_{\text{hard}}(\alpha, \boldsymbol{\mu}, \boldsymbol{\rho}; \gamma) = \alpha \cdot \mathbb{1}\{\frac{|\boldsymbol{\mu}|}{\boldsymbol{\sigma}} < \gamma\} = \alpha \cdot \mathbb{1}\{\frac{|\boldsymbol{\mu}|}{\log(1 + \exp(\boldsymbol{\rho}))} < \gamma\}, \tag{10}$$

where $\gamma$ is the hyperparameter that controls the pruning rate and $\mathbb{1}$ is the indicator function. With the above soft and hard prompt construction methods, we can modulate the tuning of task-specific prompts to deal with new tasks based on the uncertainty of the posterior meta-prompt. The tuning for task-specific prompts in Eq. 1 can be reformulated as

$$\begin{cases} \boldsymbol{s}' = \boldsymbol{s} - h_{\text{soft}}(\alpha, \boldsymbol{\rho})\nabla_{\boldsymbol{s}}\mathcal{L}_\tau(\boldsymbol{s}, \boldsymbol{\theta}), & \boldsymbol{s} \sim q(\boldsymbol{s}; \boldsymbol{\phi}), \\ \boldsymbol{s}' = \boldsymbol{s} - h_{\text{hard}}(\alpha, \boldsymbol{\mu}, \boldsymbol{\rho}; \gamma)\nabla_{\boldsymbol{s}}\mathcal{L}_\tau(\boldsymbol{s}, \boldsymbol{\theta}), & \boldsymbol{s} \sim q(\boldsymbol{s}; \boldsymbol{\phi}), \end{cases} \tag{11}$$

which are for the soft and hard construction for tasks-specific prompts, respectively.

## 5 EXPERIMENT

In what follows, we focus on three research problems: (1) What is the effectiveness and efficiency of the meta-knowledge extraction method (i.e., UAPML), especially compared to the meta-knowledge adaption methods? (2) How does the learnable Bayesian meta-prompt affect knowledge extraction? (3) Can our uncertainty-aware construction for task-specific prompts capture the task-specific information? To answer the above research questions, we compare the performance on the few-shot classification of our proposed UAPML and the following state-of-the-art baselines: (1) MAML (Finn et al., 2017): a classic meta-knowledge adaption method. (2) ANIL (Raghu et al., 2019): a more efficient method based on MAML but only adapts the classifier to tasks. (3) BOIL (Oh et al., 2020): a method instead adapts the backbone of the model during the inner adaption. (4) ProtoNet (Snell et al., 2017): a classic embedding-based meta-learning method. (5) Meta-SGD (Li et al., 2017): an efficient method that additionally learns the learning rate to adjust the one-step inner adaption. (6) Sparse-MAML (Von Oswald et al., 2021): a state-of-the-art method aims to learn which weight to change in the inner adaption. (7) LEO (Rusu et al., 2019): a conditional meta-learning method that generates a task-specific classification head. (8) Sharp-MAML (Abbas et al., 2022): a state-of-the-art variant of MAML that employs sharpness-aware minimization to achieve better performance.

In our experiment, the classification tasks consist of four sub-datasets: Aircraft (Maji et al., 2013), CIFAR-FS (Bertinetto et al., 2018), Mini-Imagenet (Vinyals et al., 2016) and miniQuickDraw (Ha & Eck, 2017). This mixture task distribution contains the classification of grey-scale images, fine-grained images and coarse-grained images. We evaluate the performance of models using the 5-way 1-shot and 5-way 5-shot settings. For meta-training, tasks are randomly sampled from the four datasets, with each task containing five randomly selected classes. After training, the performance is evaluated on four datasets respectively. Moreover, we also conduct experiments on the vanilla datasets, Mini-Imagenet and Tiered-Imagenet (Ren et al., 2018) respectively, to confirm the effectiveness and efficiency of UAPML. More details about the experiment are shown in Appendix A.4.

### 5.1 OVERALL PERFORMANCE

**RQ1: The Overall Comparision**  As shown in in Tab. 1 and Tab. 2, our proposed UAPML, including UAPML$_{soft}$ and UAPML$_{hard}$, exhibits improved performance when trained with both Conv4 and ResNet12 architecture under both 5-shot and 1-shot settings. Specifically, compared to the meta-knowledge adaption methods (i.e., MAML, ANIL and BOIL), UAPML achieve a noticeable improvement of 3.5% and 5.5% under 5-shot and 1-shot setting when using the Conv4 architecture. Likewise, when employing a larger ResNet12 architecture, UAPML achieves an improvement of 1.6% and 3.4% under the 5-shot and 1-shot settings, respectively. Moreover, our proposed UAPML save a 58% and 29% when training the Conv4 architecture under 5-shot and 1-shot settings respectively, and such a time-saving increase to 76% and 55% when training the ResNet12 architecture. This confirms the effectiveness and efficiency of meta-knowledge extraction paradigm.

In terms of efficiency, while the embedding-based method ProtoNet requires the least computational resources, it demonstrates a substantial performance gap when compared to UAPML, especially under the 1-shot setting. Comparatively, our proposed UAPML requires similar computational resources as other efficient meta-learning methods such as Meta-SGD and Sparse-MAML,

Table 1: The 5-way **5-shot** comparison of average accuracy and the time consumption on 4-layer convolution network and 12-layer ResNet.

| | Conv4 | | | ResNet12 | | |
|---|---|---|---|---|---|---|
| | Average Accuracy | Training Time[†] | Adaption Time[†] | Average Accuracy | Training Time[†] | Adaption Time[†] |
| MAML | $68.42_{\pm 0.61}$ | 1.00 | 1.00 | $72.62_{\pm 0.60}$ | 1.00 | 1.00 |
| ANIL | $68.40_{\pm 0.59}$ | 0.40 | 0.58 | $72.33_{\pm 0.61}$ | 0.22 | 0.34 |
| BOIL | $66.77_{\pm 0.60}$ | 0.99 | 0.98 | $48.01_{\pm 0.57}$ | 1.00 | 0.99 |
| ProtoNet | $64.74_{\pm 0.60}$ | 0.30 | 0.14 | $72.78_{\pm 0.54}$ | 0.15 | 0.10 |
| Meta-SGD | $58.83_{\pm 0.61}$ | 0.41 | 0.34 | $53.88_{\pm 0.59}$ | 0.31 | 0.30 |
| Sparse-MAML | $65.10_{\pm 0.65}$ | 0.43 | 0.93 | $73.02_{\pm 0.55}$ | 0.24 | 2.09 |
| LEO | $68.44_{\pm 0.60}$ | 0.81 | 3.98 | $73.02_{\pm 0.55}$ | 0.68 | 7.74 |
| Sharp-MAML | $66.61_{\pm 0.58}$ | 3.28 | 2.27 | $73.49_{\pm 0.58}$ | 4.16 | 4.91 |
| UAPML$_{soft}$ | $\mathbf{71.96_{\pm 0.59}}$ | 0.42 | 0.67 | $73.77_{\pm 0.57}$ | 0.24 | 0.39 |
| UAPML$_{hard}$ | $71.81_{\pm 0.57}$ | 0.42 | 0.66 | $\mathbf{74.25_{\pm 0.56}}$ | 0.24 | 0.39 |

[†] The time is normalized to MAML time.

Table 2: The 5-way **1-shot** comparison of average accuracy and the time consumption on 4-layer convolution network and 12-layer ResNet.

| | Conv4 | | | ResNet12 | | |
|---|---|---|---|---|---|---|
| | Average Accuracy | Training Time[†] | Adaption Time[†] | Average Accuracy | Training Time[†] | Adaption Time[†] |
| MAML | $49.16_{\pm 0.67}$ | 1.00 | 1.00 | $55.11_{\pm 0.74}$ | 1.00 | 1.00 |
| ANIL | $51.09_{\pm 0.69}$ | 0.68 | 0.60 | $54.44_{\pm 0.74}$ | 0.41 | 0.37 |
| BOIL | $46.58_{\pm 0.67}$ | 0.98 | 0.97 | $34.25_{\pm 0.57}$ | 1.00 | 1.00 |
| ProtoNet | $43.34_{\pm 0.61}$ | 0.51 | 0.12 | $54.82_{\pm 0.69}$ | 0.35 | 0.10 |
| Meta-SGD | $42.47_{\pm 0.62}$ | 0.62 | 0.35 | $41.71_{\pm 0.64}$ | 0.47 | 0.31 |
| Sparse-MAML | $45.67_{\pm 0.68}$ | 0.73 | 0.93 | $56.60_{\pm 0.76}$ | 0.47 | 1.22 |
| LEO | $51.09_{\pm 0.67}$ | 1.41 | 4.08 | $56.87_{\pm 0.80}$ | 0.91 | 3.43 |
| Sharp-MAML | $49.42_{\pm 0.68}$ | 2.83 | 2.18 | $56.60_{\pm 0.76}$ | 3.86 | 2.94 |
| UAPML$_{soft}$ | $\mathbf{56.84_{\pm 0.73}}$ | 0.71 | 0.61 | $\mathbf{58.52_{\pm 0.77}}$ | 0.45 | 0.38 |
| UAPML$_{hard}$ | $56.36_{\pm 0.73}$ | 0.71 | 0.61 | $57.56_{\pm 0.77}$ | 0.45 | 0.38 |

[†] The time is normalized to MAML time.

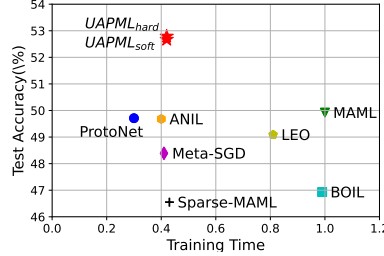
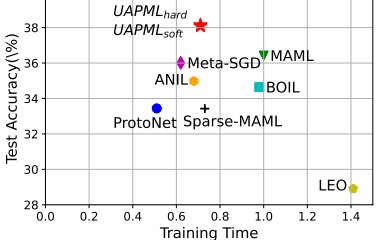

(a) 5-way 5-shot on Tiered-Imagenet  (b) 5-way 1-shot on Tiered-Imagenet

Figure 3: The 5-way 5-shot and 5-way 1-shot accuracy and training time of on Tiered-Imagenet.

but achieves superior performance. Furthermore, when compared to state-of-the-art methods such as LEO and Sharp-MAML, UAPML demonstrates significant improvements in performance across all settings, with the exception of the 5-shot evaluation on ResNet12 where UAPML still exhibits a marginal increase. This further confirms the efficiency and effectiveness of UAPML. More details about the test evaluation on each sub-dataset can be seen in Appendix A.5.

Furthermore, we provide results for all methods in Fig. 3, excluding Sharp-MAML due to its high computational requirements, on the vanilla datasets Tiered-Imagenet. The results on the Mini-Imagenet dataset follow a similar trend and are included in the Appendix A.5. The shown results illustrate that our UAPML still outperforms most of the baselines and even achieves a comparable performance compared to MAML which requires a much larger consumption. This confirms that UAPML would be the best choice when considering both effectiveness and efficiency.

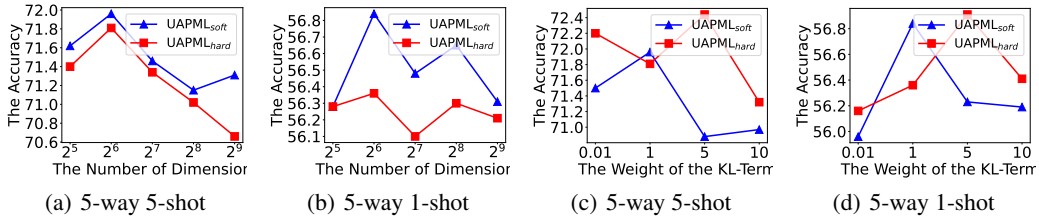

| (a) 5-way 5-shot | (b) 5-way 1-shot | (c) 5-way 5-shot | (d) 5-way 1-shot |

Figure 4: The average performance on four datasets with different dimensions (**left**) and with different weight of the KL-term (**right**) under 5-shot and 1-shot settings.

Table 3: The result of the ablation study.

|        | $UAPML_{soft}$ | $UAPML_{hard}$ | w/o Bayes | w/o C |
|--------|----------------|----------------|-----------|-------|
| 5-shot | $71.96_{\pm 0.59}$ | $71.81_{\pm 0.57}$ | $70.53_{\pm 0.60}(\downarrow)$ | $71.13_{\pm 0.58}(\downarrow)$ |
| 1-shot | $56.84_{\pm 0.73}$ | $56.36_{\pm 0.73}$ | $55.47_{\pm 0.74}(\downarrow)$ | $55.82_{\pm 0.73}(\downarrow)$ |

## 5.2 THE AFFECT OF THE PROPOSED PROMPT

**RQ2: The affect of Bayesian Meta-Prompt**   To further investigate the effectiveness of our proposed Bayesian meta-prompt, we conducted experiments with varying hyper-parameters to explore its design. We vary the number of the dimension of the meta-prompt from $2^5$ to $2^9$. The results, depicted in Fig. 4, indicate that under the 5-shot setting, a smaller dimension (e.g., from $2^6$ to $2^5$) led to a nearly 0.5% decrease in performance, while increasing the dimension resulted in a significant reduction of more than 1%. Similar patterns were observed under the 1-shot setting, where a smaller dimension negatively impacted performance, and a larger dimension did not yield performance improvements and could even have a detrimental effect. The weight assigned to the KL-term (see Eq. 4) in Bayesian inference played a crucial role, denoting the importance of the uncertainty measure. Following prior research (Higgins et al., 2017), we examined the impact of enlarging and reducing the weight of the KL-term on the performance of our approach (UAPML). As illustrated in Fig. 4, a smaller weight for the KL-term resulted in worse performance, except for $UAPML_{hard}$. On the other hand, a larger weight improved the performance of both $UAPML_{soft}$ and $UAPML_{hard}$, although excessively large weights (e.g., weight=10) had a negative effect. These findings confirm that the Bayesian treatment effectively captures task uncertainty to enhance performance, and incentivizing the capture of uncertainty in the meta-prompt is beneficial for hard construction. Additionally, we conducted an ablation study by removing the Bayesian treatment (i.e., w/o Bayes in Tab. 3), which led to a decrease in average performance ranging from nearly 1% to 1.5%. This further supports the effectiveness of the Bayesian treatment on the meta-prompt.

**RQ3: The Construction of Task-Specific Prompts**   We further conduct experiments to explore our proposed uncertainty-aware constructions for task-specific prompts. To achieve this aim, we first evaluate the variant of our methods, which constructs task-specific prompts without any modulation (i.e., w/o C). The results shown in Tab. 3 illustrate that there is a reduction of nearly 1% on the way of construction without any uncertainty-aware modulation. It confirms that the uncertainty-aware construction for task-specific prompts helps to improve the performance on new tasks.

## 6 CONCLUSION

This paper introduces UAPML, an efficient meta-learning method that employs meta-knowledge extraction instead of conventional meta-knowledge adaptation. The method incorporates a task-specific prompt to facilitate efficient utilization of acquired meta-knowledge and utilizes a learnable Bayesian meta-prompt to enable the quick adaption of task-specific prompts on new tasks. Furthermore, by analyzing the posterior uncertainty of the meta-prompt, we propose soft and hard modulation techniques to construct task-specific prompts, striking a balance between capturing task-specific information and maintaining shared knowledge. Experimental results demonstrate that UAPML outperforms other meta-learning methods even with less consumption, highlighting the effectiveness and efficiency of the meta-knowledge extraction approach.

UAPML significantly contributes to the progress of meta-learning by offering an efficient approach to leveraging meta-knowledge and establishing a more effective and efficient framework for few-shot learning. This research opens up possibilities for further exploration and improvement in the field of meta-learning and paves the way for developing more powerful and flexible learning systems.

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

# A    APPENDIX

## A.1    THE ELBO OF THE META-PROMPT

Given that we are employing the Bayesian approach to analyze the meta-prompt, it is essential to obtain its posterior distribution. However, obtaining the exact posterior distribution $p(s|\mathcal{D})$ of the meta-prompt is infeasible. In order to address this issue, we resort to the variational inference method (Kingma & Welling, 2013) to derive an approximate distribution. Specifically, we employ a variational distribution $q(s; \phi)$ as an approximation to the true posterior. This variational distribution can be learned by maximizing the log-likelihood of all datasets $\log p(\mathcal{D}; \theta)$, which can be achieved by maximizing the lower bound on the evidence (ELBO). The ELBO of the log-likelihood can be obtained through the application of Jensen's inequality:

$$
\begin{aligned}
\log p(\mathcal{D}; \theta) &= \log \int p(\mathcal{D}, s; \theta) ds \\
&= \log \int p(\mathcal{D}|s; \theta) p(s) ds \\
&= \log \int \frac{q(s; \phi) p(\mathcal{D}|s; \theta) p(s)}{q(s; \phi)} ds \\
&\geq \mathbb{E}_{q(s; \phi)} \left[ \log p(\mathcal{D}|s; \theta) \right] - \mathbf{KL} \left[ q(s; \phi) \| p(s) \right],
\end{aligned}
\tag{12}
$$

where KL denotes the Kullback–Leibler divergence that is utilized as a measure of the dissimilarity between two probability distributions. In Equation 12, the first term represents the log-likelihood of all datasets conditioned on the variational distribution of the meta-prompt. The second term corresponds to the KL divergence between the variational posterior and the prior distribution of the meta-prompt.

Note that we assume $q(s; \phi)$ to be a Gaussian with diagonal covariance, which is parameterized by $\phi = \mu, \rho$. The standard deviation is parameterized as $\sigma = \log(1 + \exp(\rho))$ to avoid the invalid value (i.e., the negative value). For the prior distribution of meta-prompts, we follow (Blundell et al., 2015) to use a mixture of Gaussian distribution, which consists of one standard Gaussian and another zero-centered Gaussian but with a larger variance to provide a heavier tail in the prior density. Since such a mixture prior focuses more on the tail part, it prevents the posterior from collapsing into some extreme points, leading to improved performance. We compute the ELBO in Eq. 4 using Monte Carlo sampling with the reparameterization trick (Kingma & Welling, 2013), where we sample from variational distributions to approximate the expectation of log-likelihood as well as the KL-term.

## A.2    THE DETAILED PROOF FOR THE PROPOSITION

In this subsection, we present a detailed derivation of the proof for Proposition 1. We adopt a different perspective to decouple the objective as follows:

$$
\begin{aligned}
\mathcal{L}_{LB}(\theta, \phi) &= - \mathbf{KL} \left[ q(\mathcal{D}, s', s; \theta, \phi) \| p(\mathcal{D}, s', s) \right] \\
&= \mathbb{E}_{q(\mathcal{D}, s', s; \theta, \phi)} \left[ \log \frac{p(\mathcal{D}|s', s) p(s'|s) p(s) q(s', s; \theta, \phi) p(\mathcal{D})}{q(s', s|\mathcal{D}; \theta, \phi) q(\mathcal{D}) q(s', s; \theta, \phi) p(\mathcal{D})} \right] \\
&= \mathbb{E}_{q(\mathcal{D}, s', s; \theta, \phi)} \left[ \log \frac{p(\mathcal{D}|s', s)}{p(\mathcal{D})} - \log \frac{q(s', s|\mathcal{D}; \theta, \phi)}{q(s', s; \theta, \phi)} - \log \frac{q(\mathcal{D})}{p(\mathcal{D})} - \log \frac{q(s'|s; \theta)}{p(s'|s)} - \log \frac{q(s; \phi)}{p(s)} \right] \\
&= \mathbb{E}_{q(\mathcal{D}, s', s; \theta, \phi)} \left[ \log \frac{p(\mathcal{D}|s', s)}{p(\mathcal{D})} - \log \frac{q(s', s|\mathcal{D}; \theta, \phi)}{q(s', s; \theta, \phi)} \right] \\
&\quad - \mathbf{KL} \left[ q(\mathcal{D}) \| p(\mathcal{D}) \right] - \mathbf{KL} \left[ q(s'|s; \theta) \| p(s'|s) \right] - \mathbf{KL} \left[ q(s; \phi) \| p(s) \right],
\end{aligned}
$$

where the first term comprises of two components - the log-likelihood of all the datasets conditioned on the prompt and the ratio between the meta-prompt and task-specific prompt conditioned and unconditioned on all the datasets. The remaining three terms in the objective are KL-divergences between the variational distributions and the priors. In particular, we emphasize the latter part of the first term, which can be interpreted as the mutual information between the joint distribution of the

meta-prompt and task-specific prompts and all the datasets. This mutual information can be further derived as:

$$
\begin{aligned}
I(\boldsymbol{s}', \boldsymbol{s}; \mathcal{D}) &= \mathbb{E}_{q(\boldsymbol{s}',\boldsymbol{s},\mathcal{D})} \left[ \log \frac{q(\boldsymbol{s}', \boldsymbol{s}|\mathcal{D}; \boldsymbol{\theta}, \boldsymbol{\phi})}{q(\boldsymbol{s}', \boldsymbol{s}; \boldsymbol{\theta}, \boldsymbol{\phi})} \right] \\
&= \mathbb{E}_{q(\boldsymbol{s}',\boldsymbol{s},\mathcal{D})} \left[ \log \frac{q(\boldsymbol{s}'|\boldsymbol{s}, \mathcal{D}; \boldsymbol{\theta}) q(\boldsymbol{s}; \boldsymbol{\phi})}{q(\boldsymbol{s}'|\boldsymbol{s}; \boldsymbol{\theta}) q(\boldsymbol{s}; \boldsymbol{\phi})} \right] \\
&= \mathbb{E}_{q(\boldsymbol{s}',\boldsymbol{s},\mathcal{D})} \left[ q(\boldsymbol{s}'|\boldsymbol{s}, \mathcal{D}; \boldsymbol{\theta}) \right] - \mathbb{E}_{q(\boldsymbol{s}',\boldsymbol{s},\mathcal{D})} \left[ q(\boldsymbol{s}'|\boldsymbol{s}; \boldsymbol{\theta}) \right] \\
&= H(q(\boldsymbol{s}'|\boldsymbol{s}, \mathcal{D}; \boldsymbol{\theta})) - H(q(\boldsymbol{s}'|\boldsymbol{s}; \boldsymbol{\theta})) \\
&\approx H(q(\boldsymbol{s}'|\boldsymbol{s}, \mathcal{D}; \boldsymbol{\theta})) - H(q(\boldsymbol{s}; \boldsymbol{\phi})),
\end{aligned}
\tag{13}
$$

where the mutual information term can be separated into the difference between two information entropy. In the absence of datasets, the task-specific prompts retain their initializations that are drawn from the distribution of the meta-prompt, thereby supporting the above derivation. Consequently, one of the goals is to align the information entropy of the meta-prompt with that of the task-specific prompts.

---

**Algorithm 1:** The meta-train stage of UAPML.

---

**Input:** the meta-training datasets $p(\mathcal{D})$, the learning rate $\beta$ and $\alpha$ for the outer loop and inner loop, the number of inner steps $n$.
**Output:** the meta-parameters $\boldsymbol{\theta}_r$, $\boldsymbol{\theta}_m$ and $\boldsymbol{\theta}_c$, the posterior of the meta-prompt $q(\boldsymbol{s})$ parameterized by $\boldsymbol{\phi}$.
1: Initialize meta-parameters and the meta-prompt: $\boldsymbol{\theta}_r$, $\boldsymbol{\theta}_m$, $\boldsymbol{\theta}_c$ and $q(\boldsymbol{s}; \boldsymbol{\phi})$;
2: **while** not converage **do**
3:     Sample a tasks $\mathcal{D} \sim p(\mathcal{D})$;
4:     Sample a meta-prompt $\boldsymbol{s} \sim q(\boldsymbol{s}; \boldsymbol{\phi})$;
5:     Initialize the task-specific prompts with the sampled meta-prompt: $\boldsymbol{s}' \leftarrow \boldsymbol{s}$;
6:     **for** i = 1,2,..,n **do**
7:         Compute the Loss $\mathcal{L}(\boldsymbol{s}, \boldsymbol{\theta})$;
8:         Update the task-specific prompt $\boldsymbol{s}'$ according to Eq. 1;
9:         Update the task-specific classification $\boldsymbol{\theta}_c$ with the learning rate $\alpha$;
10:     **end for**
11:     Compute the ELBO according to Eq. 4;
12:     Update $\boldsymbol{\theta}_r$, $\boldsymbol{\theta}_m$, $\boldsymbol{\theta}_r$ and the variational parameters $\boldsymbol{\phi}$ of the meta-prompt with the learning rate $\beta$;
13: **end while**

---

**Algorithm 2:** The meta-test stage of UAPML

---

**Input:** the meta-parameters $\boldsymbol{\theta}_r$, $\boldsymbol{\theta}_m$ and $\boldsymbol{\theta}_c$, the posterior meta-prompt $q(\boldsymbol{s})$ parameterized by $\boldsymbol{\phi} = \{\boldsymbol{\mu}, \boldsymbol{\rho}\}$, the tuning rate $\alpha$, the number of inner steps $n$ and the task $\tau$.
**Output:** the task-specific prompts $\boldsymbol{s}'$ and the task-specific head $\boldsymbol{\theta}'_c$.
1: Sample an initialization for task-specific prompts from the posterior meta-prompt:
    $\boldsymbol{s}' \leftarrow \boldsymbol{s}, \ \boldsymbol{s} \sim q(\boldsymbol{s}; \boldsymbol{\phi})$;
2: **for** $i = 1, 2, 3, ..., n$ **do**
3:     Compute the tuning rate $h(\alpha)$ according to Eq. 9 or Eq. 10;
4:     Tune the task-specific prompt $\boldsymbol{s}'$ and the task-specific head according to Eq. 11 and Eq. 3;
5: **end for**

---

### A.3 THE ANALYSIS OF THE TIME COMPLEXITY

The pseudo-code of the meta-train and meta-test stage of our proposed UAPML are shown in Alg.1 and Alg.2, respectively. One of the benefits of the meta-knowledge extraction paradigm is its efficiency. Hence, we analyze the time complexity of several popular meta-learning methods, compared to our proposed UAPML.

MAML (Finn et al., 2017), the most widely used meta-learning approach, employs a bi-level optimization scheme that involves an outer loop for updating meta-parameters and an inner loop for evaluating the current meta-parameters. Consequently, its time complexity is $\mathcal{O}(n^2)$ due to this bi-level optimization architecture. BOIL (Oh et al., 2020), like MAML, freezes only the classifier on the last layer and still updates most of the model parameters during the inner adaptation, resulting in a time complexity of $\mathcal{O}(n^2)$. ANIL (Raghu et al., 2019), on the other hand, updates only the last layer, i.e., the classifier, during the inner adaptation, leading to a reduced time complexity of $\mathcal{O}(n)$. Similarly, our proposed UAPML only tunes the last layer and a small-scale prompt, both of which are located on the last layers, resulting in significant computation reduction for gradients. Therefore, the time complexity of our proposed UAPML is similar to ANIL, i.e., $\mathcal{O}(n)$. The experimental results in the following section demonstrate the effectiveness of our proposed method in comparison to these state-of-the-art meta-learning approaches.

### A.4 THE DETAILS OF EXPERIMENT

#### A.4.1 THE DATASETS

In what follows, we present a detailed introduction of the datasets used in our experiments:

**Aircraft**: Aircraft (Maji et al., 2013) is a fine-grained dataset containing 100 aircraft model classes, each with 100 images. We randomly divide 64 classes for training, 16 classes for validation, and 20 classes for evaluation.

**CIFAR-FS**: CIFAR-FS (Bertinetto et al., 2018) is a well-known image classification dataset comprising 100 classes of objects, each with 600 images. We randomly select 100 images for each class and randomly divide 64 classes for training, 16 classes for validation, and 20 classes for evaluation.

**miniImageNet**: miniImageNet (Vinyals et al., 2016) is a popular dataset for meta-learning, which consists of 100 classes and 600 images for each class from the ImageNet dataset. We randomly select 100 images for each class and split 100 classes into 64 classes for training, 16 classes for validation, and 20 classes for evaluation.

**miniQuickDraw**: miniQuickDraw (Ha & Eck, 2017) is a grayscale image classification dataset containing 345 classes sampled from QuickDraw. Consistent with the other datasets, we randomly select 100 classes with each class containing 100 images. The selected classes are split into 64 classes for training, 16 classes for validation, and the remaining 20 classes for evaluation.

**Tiered-Imagenet** Tiered-Imagenet (Ren et al., 2018) is a larger subset of ILSVRC-12 (608 classes rather than 100 for miniImageNet). We randomly select 1300 images for each class, and split 608 classes into 351 classes for training, 97 classes for validation and the remaining 160 classes for evaluation.

#### A.4.2 THE BASELINES

To evaluate the effectiveness of our proposed UAPML and address the research problems, we compare our method with the following meta-learning approaches:

**MAML** (Finn et al., 2017): a classic method that learns a good initialization for model parameters and adapts fully to task-specific parameters, requiring a bi-level optimization architecture with an outer loop for updating the initialization and an inner loop for evaluating the initialization.

**ANIL** (Raghu et al., 2019): a method based on MAML that proposes to use feature reuse instead of quick adaptation to reduce computational overheads. It freezes the model backbone (i.e., feature extractor) and only updates the task head when dealing with few-shot tasks.

**BOIL** (Oh et al., 2020): a method that adapts the backbone of the model during the inner adaptation to handle different task distributions, so as to provide different features for tasks, which is in contrast to ANIL.

**ProtoNet** (Snell et al., 2017): a classic embedding-based meta-learning method that learns a metric space for few-shot tasks. Since it does not need the inner adaption, the computational consumption is much less compared to MAML.

Table 4: **5-way 5-shot** accuracy (%) with 95% confidence interval of Conv4 architecture on each task distribution and on average. The best and the second-best performances are bolded and underlined, respectively.

| | Aircraft | CIFAR-FS | Mini-Imagenet | miniQuickDraw | Average |
|---|---|---|---|---|---|
| MAML | $69.00_{\pm 0.59}$ | $67.71_{\pm 0.64}$ | $59.27_{\pm 0.69}$ | $\mathbf{77.68_{\pm 0.51}}$ | $68.42_{\pm 0.61}$ |
| ProtoNet | $64.06_{\pm 0.60}$ | $63.21_{\pm 0.62}$ | $57.03_{\pm 0.64}$ | $74.64_{\pm 0.54}$ | $64.74_{\pm 0.60}$ |
| ANIL | $70.11_{\pm 0.59}$ | $67.93_{\pm 0.60}$ | $59.08_{\pm 0.65}$ | $76.48_{\pm 0.53}$ | $68.40_{\pm 0.59}$ |
| BOIL | $68.20_{\pm 0.57}$ | $65.98_{\pm 0.62}$ | $58.82_{\pm 0.66}$ | $74.10_{\pm 0.54}$ | $66.77_{\pm 0.60}$ |
| Meta-SGD | $55.94_{\pm 0.58}$ | $57.05_{\pm 0.63}$ | $52.95_{\pm 0.63}$ | $69.39_{\pm 0.58}$ | $58.83_{\pm 0.61}$ |
| LEO | $70.10_{\pm 0.59}$ | $68.35_{\pm 0.59}$ | $58.52_{\pm 0.62}$ | $76.76_{\pm 0.60}$ | $68.44_{\pm 0.60}$ |
| Sparse-MAML | $67.98_{\pm 0.61}$ | $63.89_{\pm 0.68}$ | $56.14_{\pm 0.70}$ | $72.40_{\pm 0.62}$ | $65.10_{\pm 0.65}$ |
| Sharp-MAML | $68.06_{\pm 0.57}$ | $66.43_{\pm 0.60}$ | $56.34_{\pm 0.64}$ | $75.62_{\pm 0.53}$ | $66.61_{\pm 0.58}$ |
| UAPML$_{soft}$ | $76.32_{\pm 0.54}$ | $\mathbf{72.37_{\pm 0.60}}$ | $\mathbf{61.83_{\pm 0.65}}$ | $\underline{77.31_{\pm 0.55}}$ | $\mathbf{71.96_{\pm 0.59}}$ |
| UAPML$_{hard}$ | $\mathbf{76.53_{\pm 0.56}}$ | $71.48_{\pm 0.56}$ | $61.59_{\pm 0.65}$ | $77.63_{\pm 0.52}$ | $71.81_{\pm 0.57}$ |

Table 5: **5-way 1-shot** accuracy (%) with 95% confidence interval of Conv4 architecture on each task distribution and on average. The best and the second-best performances are bolded and underlined, respectively.

| | Aircraft | CIFAR-FS | Mini-Imagenet | miniQuickDraw | Average |
|---|---|---|---|---|---|
| MAML | $47.80_{\pm 0.61}$ | $48.18_{\pm 0.70}$ | $42.17_{\pm 0.65}$ | $58.49_{\pm 0.71}$ | $49.16_{\pm 0.67}$ |
| ProtoNet | $34.70_{\pm 0.51}$ | $44.49_{\pm 0.65}$ | $39.23_{\pm 0.60}$ | $54.94_{\pm 0.69}$ | $43.34_{\pm 0.61}$ |
| ANIL | $53.71_{\pm 0.65}$ | $48.88_{\pm 0.70}$ | $44.03_{\pm 0.69}$ | $57.72_{\pm 0.72}$ | $51.09_{\pm 0.69}$ |
| BOIL | $40.27_{\pm 0.58}$ | $47.18_{\pm 0.69}$ | $41.71_{\pm 0.68}$ | $57.15_{\pm 0.74}$ | $46.58_{\pm 0.67}$ |
| Meta-SGD | $39.10_{\pm 0.56}$ | $40.89_{\pm 0.65}$ | $37.85_{\pm 0.61}$ | $53.07_{\pm 0.65}$ | $42.47_{\pm 0.62}$ |
| LEO | $53.95_{\pm 0.65}$ | $47.15_{\pm 0.65}$ | $41.49_{\pm 0.63}$ | $\mathbf{61.78_{\pm 0.75}}$ | $51.09_{\pm 0.67}$ |
| Sparse-MAML | $47.95_{\pm 0.66}$ | $43.00_{\pm 0.72}$ | $38.68_{\pm 0.66}$ | $53.05_{\pm 0.70}$ | $45.67_{\pm 0.68}$ |
| Sharp-MAML | $51.30_{\pm 0.68}$ | $46.69_{\pm 0.67}$ | $41.62_{\pm 0.65}$ | $58.08_{\pm 0.70}$ | $49.42_{\pm 0.68}$ |
| UAPML$_{soft}$ | $\mathbf{64.27_{\pm 0.75}}$ | $\mathbf{54.73_{\pm 0.75}}$ | $\mathbf{46.91_{\pm 0.71}}$ | $\underline{61.47_{\pm 0.71}}$ | $\mathbf{56.84_{\pm 0.73}}$ |
| UAPML$_{hard}$ | $\underline{63.12_{\pm 0.75}}$ | $\underline{54.11_{\pm 0.73}}$ | $\underline{46.85_{\pm 0.73}}$ | $61.34_{\pm 0.72}$ | $\underline{56.36_{\pm 0.73}}$ |

**Meta-SGD** (Li et al., 2017): an efficient method that learns the appropriate learning rate to enable the one-step adaption to few-shot tasks.

**Sparse-MAML** (Von Oswald et al., 2021): a state-of-the-art efficient method that aims to learn which weight to change during inner adaptation, allowing apart of parameters to adapt to tasks. It employs an additional modulator with the same scale as the model architecture to determine where to update during inner adaptation.

**LEO** (Rusu et al., 2019): a conditional meta-learning method that captures task information and operates the inner adaptation in a latent space. It then uses the latent vector to generate a task-specific task head to deal with the specific task.

**Sharp-MAML** (Abbas et al., 2022): a state-of-the-art variant of MAML that employs sharpness-aware minimization tackle the complex loss landscape caused by the bi-level optimization, achieving a better performance.

### A.4.3 THE DETAILS OF THE EXPERIMENTAL SETTINGS

We employ the same architecture as the base model in MAML (Finn et al., 2017) for all the baselines and our proposed model. The architecture comprises 4 convolution layers with 64 filters, batch normalization, a ReLU non-linear layer, and a $2 \times 2$ max-pool. A larger architecture ResNet12 is also used to examinate the effectiveness and efficiency of all methpds. Except for the conventional methods (i.e., MAML, ANIL, and BOIL), all the baselines are based on the public repositories released by the original papers. In our proposed UAPML, we use the full-connect layer as the merge module, which is the simplest layer. We set the tuning step as 4 and the tuning rate as 0.01 in UAPML. Moreover, we use Adam optimization with a learning rate of 0.001 to learn the meta-prompt and other meta-parameters. All models are trained on NVIDIA A40 for 10000 epochs, with each epoch consisting of a batch of 4 tasks from different task distributions. The training process takes less than one hour under both 5-shot and 1-shot settings.

Table 6: **5-way 5-shot** accuracy (%) with 95% confidence interval of ResNet12 architecture on each task distribution and on average. The best and the second-best performances are bolded and underlined, respectively.

| | Aircraft | CIFAR-FS | Mini-Imagenet | miniQuickDraw | Average |
|---|---|---|---|---|---|
| MAML | $76.46_{\pm0.58}$ | $72.04_{\pm0.62}$ | $62.08_{\pm0.68}$ | $\underline{79.91_{\pm0.51}}$ | $72.62_{\pm0.60}$ |
| ProtoNet | $76.35_{\pm0.50}$ | $71.60_{\pm0.59}$ | $61.91_{\pm0.61}$ | $\mathbf{81.25_{\pm0.46}}$ | $72.78_{\pm0.54}$ |
| ANIL | $76.87_{\pm0.59}$ | $72.32_{\pm0.60}$ | $62.38_{\pm0.66}$ | $77.77_{\pm0.58}$ | $72.33_{\pm0.61}$ |
| BOIL | $32.85_{\pm0.51}$ | $49.38_{\pm0.60}$ | $45.49_{\pm0.60}$ | $64.31_{\pm0.58}$ | $48.01_{\pm0.57}$ |
| Meta-SGD | $60.89_{\pm0.56}$ | $51.27_{\pm0.58}$ | $46.48_{\pm0.62}$ | $56.87_{\pm0.60}$ | $53.88_{\pm0.59}$ |
| LEO | $76.30_{\pm0.55}$ | $72.06_{\pm0.57}$ | $63.86_{\pm0.62}$ | $79.85_{\pm0.47}$ | $73.02_{\pm0.55}$ |
| Sparse-MAML | $77.79_{\pm0.60}$ | $\mathbf{75.18_{\pm0.63}}$ | $\underline{63.32_{\pm0.70}}$ | $76.70_{\pm0.62}$ | $73.25_{\pm0.64}$ |
| Sharp-MAML | $77.37_{\pm0.56}$ | $74.15_{\pm0.60}$ | $63.17_{\pm0.67}$ | $79.27_{\pm0.50}$ | $73.49_{\pm0.58}$ |
| UAPML$_{soft}$ | $\mathbf{80.57_{\pm0.51}}$ | $73.66_{\pm0.60}$ | $62.74_{\pm0.68}$ | $78.11_{\pm0.51}$ | $\underline{73.77_{\pm0.57}}$ |
| UAPML$_{hard}$ | $\underline{78.32_{\pm0.51}}$ | $74.82_{\pm0.60}$ | $\mathbf{64.68_{\pm0.65}}$ | $79.20_{\pm0.49}$ | $\mathbf{74.25_{\pm0.56}}$ |

Table 7: **5-way 1-shot** accuracy (%) with 95% confidence interval of ResNet12 architecture on each task distribution and on average. The best and the second-best performances are bolded and underlined, respectively.

| | Aircraft | CIFAR-FS | Mini-Imagenet | miniQuickDraw | Average |
|---|---|---|---|---|---|
| MAML | $60.26_{\pm0.73}$ | $53.86_{\pm0.78}$ | $44.31_{\pm0.71}$ | $62.02_{\pm0.75}$ | $55.11_{\pm0.74}$ |
| ProtoNet | $60.56_{\pm0.71}$ | $53.14_{\pm0.70}$ | $46.26_{\pm0.67}$ | $59.31_{\pm0.70}$ | $54.82_{\pm0.69}$ |
| ANIL | $59.06_{\pm0.71}$ | $52.26_{\pm0.78}$ | $43.79_{\pm0.70}$ | $62.63_{\pm0.78}$ | $54.44_{\pm0.74}$ |
| BOIL | $26.61_{\pm0.47}$ | $35.44_{\pm0.61}$ | $32.02_{\pm0.53}$ | $42.93_{\pm0.65}$ | $34.25_{\pm0.57}$ |
| Meta-SGD | $27.98_{\pm0.45}$ | $38.23_{\pm0.64}$ | $35.56_{\pm0.62}$ | $45.40_{\pm0.71}$ | $36.80_{\pm0.60}$ |
| LEO | $30.64_{\pm0.50}$ | $42.62_{\pm0.69}$ | $37.87_{\pm0.65}$ | $55.71_{\pm0.73}$ | $41.71_{\pm0.64}$ |
| Sparse-MAML | $62.48_{\pm0.77}$ | $\underline{55.98_{\pm0.82}}$ | $\underline{47.01_{\pm0.79}}$ | $62.00_{\pm0.81}$ | $56.87_{\pm0.80}$ |
| Sharp-MAML | $62.15_{\pm0.76}$ | $54.84_{\pm0.80}$ | $44.70_{\pm0.69}$ | $\mathbf{64.70_{\pm0.79}}$ | $56.60_{\pm0.76}$ |
| UAPML$_{soft}$ | $\mathbf{64.92_{\pm0.77}}$ | $\mathbf{57.94_{\pm0.79}}$ | $\mathbf{47.05_{\pm0.73}}$ | $64.15_{\pm0.78}$ | $\mathbf{58.52_{\pm0.77}}$ |
| UAPML$_{hard}$ | $\underline{63.18_{\pm0.81}}$ | $55.94_{\pm0.78}$ | $46.76_{\pm0.73}$ | $\underline{64.36_{\pm0.74}}$ | $57.56_{\pm0.77}$ |

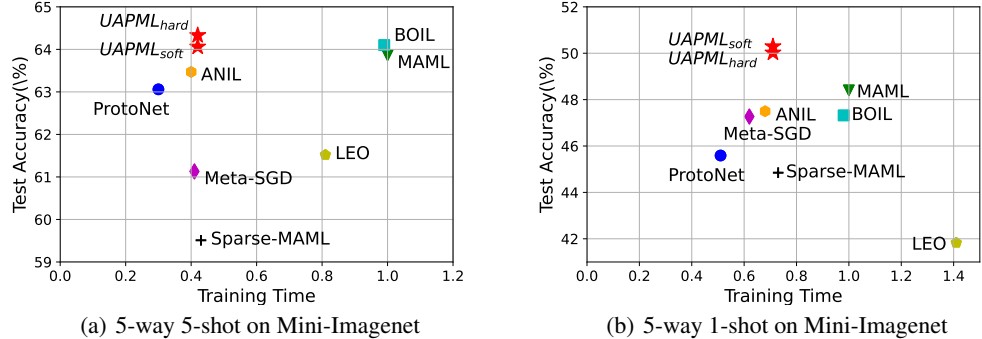

(a) 5-way 5-shot on Mini-Imagenet     (b) 5-way 1-shot on Mini-Imagenet

Figure 5: The 5-way 5-shot and 5-way 1-shot accuracy and training time on Mini-Imagenet.

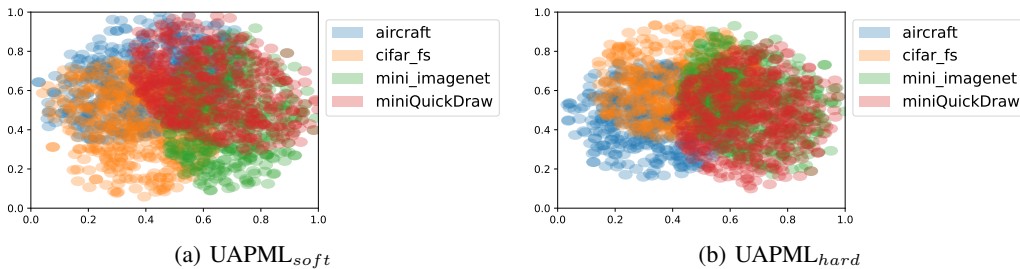

(a) UAPML$_{soft}$     (b) UAPML$_{hard}$

Figure 6: The visualization for task-specific prompts of UAPML$_{soft}$ and UAPML$_{hard}$ under 5-way 5-shot setting.

Table 8: **5-way 5-shot** accuracy (%) with 95% confidence interval of the variants of our UAPML on each task distribution and on average.

| | | Aircraft | CIFAR-FS | Mini-Imagenet | miniQuickDraw | Average |
|---|---|---|---|---|---|---|
| | dim=32 | $76.10_{\pm0.53}$ | $71.75_{\pm0.57}$ | $61.51_{\pm0.68}$ | $77.14_{\pm0.54}$ | $71.62_{\pm0.58}$ |
| | dim=64 | $76.32_{\pm0.54}$ | $72.37_{\pm0.60}$ | $61.83_{\pm0.65}$ | $77.31_{\pm0.55}$ | $71.96_{\pm0.59}$ |
| | dim=128 | $74.84_{\pm0.54}$ | $71.65_{\pm0.59}$ | $61.61_{\pm0.66}$ | $77.73_{\pm0.53}$ | $71.46_{\pm0.58}$ |
| UAPML$_{soft}$ | dim=256 | $74.19_{\pm0.55}$ | $71.80_{\pm0.60}$ | $61.42_{\pm0.66}$ | $77.19_{\pm0.53}$ | $71.15_{\pm0.59}$ |
| | dim=512 | $77.03_{\pm0.51}$ | $70.92_{\pm0.59}$ | $60.60_{\pm0.66}$ | $76.68_{\pm0.53}$ | $71.31_{\pm0.57}$ |
| | kl=0.01 | $75.13_{\pm0.57}$ | $71.71_{\pm0.59}$ | $61.50_{\pm0.67}$ | $77.67_{\pm0.51}$ | $71.50_{\pm0.59}$ |
| | kl=5 | $74.26_{\pm0.55}$ | $70.73_{\pm0.59}$ | $61.26_{\pm0.65}$ | $77.25_{\pm0.54}$ | $70.88_{\pm0.58}$ |
| | kl=10 | $73.92_{\pm0.56}$ | $71.41_{\pm0.59}$ | $61.21_{\pm0.64}$ | $77.35_{\pm0.51}$ | $70.97_{\pm0.58}$ |
| | dim=32 | $75.21_{\pm0.54}$ | $72.01_{\pm0.58}$ | $61.59_{\pm0.67}$ | $76.78_{\pm0.54}$ | $71.40_{\pm0.58}$ |
| | dim=64 | $76.53_{\pm0.56}$ | $71.48_{\pm0.56}$ | $61.59_{\pm0.65}$ | $77.63_{\pm0.52}$ | $71.81_{\pm0.57}$ |
| | dim=128 | $75.28_{\pm0.54}$ | $72.23_{\pm0.59}$ | $60.92_{\pm0.64}$ | $76.91_{\pm0.52}$ | $71.34_{\pm0.57}$ |
| UAPML$_{hard}$ | dim=256 | $75.24_{\pm0.56}$ | $70.93_{\pm0.62}$ | $61.32_{\pm0.68}$ | $76.57_{\pm0.54}$ | $71.02_{\pm0.60}$ |
| | dim=512 | $75.17_{\pm0.57}$ | $70.20_{\pm0.63}$ | $60.82_{\pm0.64}$ | $76.45_{\pm0.52}$ | $70.66_{\pm0.59}$ |
| | kl=0.01 | $76.04_{\pm0.52}$ | $72.34_{\pm0.59}$ | $62.60_{\pm0.65}$ | $77.81_{\pm0.54}$ | $72.20_{\pm0.57}$ |
| | kl=5 | $76.15_{\pm0.56}$ | $72.56_{\pm0.59}$ | $62.64_{\pm0.66}$ | $78.41_{\pm0.51}$ | $72.44_{\pm0.58}$ |
| | kl=10 | $75.94_{\pm0.56}$ | $70.61_{\pm0.57}$ | $61.82_{\pm0.66}$ | $76.91_{\pm0.53}$ | $71.32_{\pm0.58}$ |
| N/A | w/o Bayes | $74.41_{\pm0.57}$ | $69.93_{\pm0.61}$ | $60.83_{\pm0.68}$ | $76.93_{\pm0.54}$ | $70.53_{\pm0.60}$ |
| N/A | w/o C | $75.57_{\pm0.55}$ | $70.87_{\pm0.59}$ | $60.93_{\pm0.65}$ | $77.14_{\pm0.54}$ | $71.13_{\pm0.58}$ |

Table 9: **5-way 1-shot** accuracy (%) with 95% confidence interval of the variants of our UAPML on each task distribution and on average.

| | | Aircraft | CIFAR-FS | Mini-Imagenet | miniQuickDraw | Average |
|---|---|---|---|---|---|---|
| | dim=32 | $63.77_{\pm0.72}$ | $53.25_{\pm0.76}$ | $46.33_{\pm0.75}$ | $61.75_{\pm0.73}$ | $56.28_{\pm0.74}$ |
| | dim=64 | $64.27_{\pm0.75}$ | $54.73_{\pm0.75}$ | $46.91_{\pm0.71}$ | $61.47_{\pm0.71}$ | $56.84_{\pm0.73}$ |
| | dim=128 | $63.41_{\pm0.74}$ | $53.94_{\pm0.75}$ | $46.21_{\pm0.75}$ | $62.36_{\pm0.73}$ | $56.48_{\pm0.74}$ |
| UAPML$_{soft}$ | dim=256 | $62.88_{\pm0.74}$ | $54.69_{\pm0.74}$ | $47.23_{\pm0.72}$ | $61.79_{\pm0.76}$ | $56.65_{\pm0.74}$ |
| | dim=512 | $62.90_{\pm0.68}$ | $53.82_{\pm0.76}$ | $46.57_{\pm0.74}$ | $61.93_{\pm0.70}$ | $56.31_{\pm0.72}$ |
| | kl=0.01 | $62.83_{\pm0.72}$ | $52.94_{\pm0.73}$ | $47.09_{\pm0.72}$ | $60.98_{\pm0.72}$ | $55.96_{\pm0.72}$ |
| | kl=5 | $63.21_{\pm0.76}$ | $55.37_{\pm0.77}$ | $45.32_{\pm0.70}$ | $61.00_{\pm0.73}$ | $56.23_{\pm0.74}$ |
| | kl=10 | $62.80_{\pm0.74}$ | $53.49_{\pm0.77}$ | $46.36_{\pm0.71}$ | $62.10_{\pm0.75}$ | $56.19_{\pm0.74}$ |
| | dim=32 | $63.61_{\pm0.75}$ | $53.47_{\pm0.76}$ | $46.08_{\pm0.68}$ | $61.94_{\pm0.73}$ | $56.28_{\pm0.73}$ |
| | dim=64 | $63.12_{\pm0.75}$ | $54.11_{\pm0.73}$ | $46.85_{\pm0.73}$ | $61.34_{\pm0.72}$ | $56.36_{\pm0.73}$ |
| | dim=128 | $63.54_{\pm0.75}$ | $53.58_{\pm0.77}$ | $46.54_{\pm0.71}$ | $60.73_{\pm0.71}$ | $56.10_{\pm0.74}$ |
| UAPML$_{hard}$ | dim=256 | $63.51_{\pm0.74}$ | $53.77_{\pm0.73}$ | $47.09_{\pm0.73}$ | $60.83_{\pm0.77}$ | $56.30_{\pm0.74}$ |
| | dim=512 | $63.10_{\pm0.71}$ | $54.00_{\pm0.75}$ | $46.00_{\pm0.70}$ | $61.74_{\pm0.74}$ | $56.21_{\pm0.73}$ |
| | kl=0.01 | $63.51_{\pm0.73}$ | $53.38_{\pm0.75}$ | $46.04_{\pm0.70}$ | $61.71_{\pm0.75}$ | $61.71_{\pm0.75}$ |
| | kl=5 | $65.87_{\pm0.73}$ | $53.58_{\pm0.72}$ | $46.53_{\pm0.72}$ | $61.67_{\pm0.71}$ | $56.91_{\pm0.72}$ |
| | kl=10 | $63.42_{\pm0.74}$ | $52.60_{\pm0.72}$ | $46.98_{\pm0.71}$ | $62.62_{\pm0.74}$ | $56.41_{\pm0.73}$ |
| N/A | w/o Bayes | $62.07_{\pm0.76}$ | $53.07_{\pm0.75}$ | $45.27_{\pm0.72}$ | $61.47_{\pm0.73}$ | $55.47_{\pm0.74}$ |
| N/A | w/o C | $63.63_{\pm0.70}$ | $52.54_{\pm0.77}$ | $46.24_{\pm0.73}$ | $60.89_{\pm0.73}$ | $55.82_{\pm0.73}$ |

## A.5   THE EXPERIMENTAL RESULT

At first, we showed the detailed results of all methods of Conv4 architecture on the mixture datasets in Tab. 4 and Tab. 5, including the average performance and the respective performance on each dataset. The detailed results demonstrate that under both the 5-shot and 1-shot setting, UAPML achieve a statistically significant improvement on all datasets except *miniQuickDraw* according to Student's $t$-test (Sakai, 2016). This further confirms the effectiveness of our proposed UAPML. with regards to the architecture ResNet12, we also show the 5-way 5-shot and 5-way 1-shot results in Tab. 6 and Tab. 7, respectively. Although some baselines perform best on a certain single dataset, our proposed UAPML still has a significant improvement in average accuracy.

Moreover, we also report the performance of all methods on the vanilla dataset Mini-Imagenet in Fig. 5, including the test accuracy and the time consumption for training. Even on this vanilla dataset, our proposed UAPML achieves the best performance under 5-shot and 1-shot settings. Although the increase on 5-shot is not significant, our proposed UAPML only uses 40% time for training when compared to MAML. The results illustrate that our proposed UAPML is a more effective and efficient meta-learning method.

Furthermore, we show the detailed performance of the variant of UAPML, i.e., the performance on each task distribution, in this subsection. The detailed results under 5-shot and 1-shot settings are shown in Tab. 8 and Tab. 9, respectively.

We also visualize the task-specific prompt under the 5-shot setting. It is observed that task-specific prompts for each dataset are separated but still have some overlap. This is because the construction for task-specific prompts limits or even freezes part of the dimensions when adating oto tasks.

