# OpenReview forum: "Meta-Knowledge Extraction: Uncertainty-Aware Prompted Meta-Learning"
_ICLR.cc/2024/Conference — Submitted to ICLR 2024_

### Official Review · Reviewer_7FMs · 2023-10-29

**Soundness:** 2 fair
**Presentation:** 3 good
**Contribution:** 2 fair
**Rating:** 3
**Confidence:** 4

**Summary:**

A Bayesian prompt tuning approach is introduced to MAML for a few-shot image classification task. The main idea is to leverage prompt to fast adapt input to the fixed meta-leaner (feature extractor), instead of computing nested gradients on the whole model, in order to improve learning/inference efficiency. Experiments on four benchmark datasets were provided under two simple backbone models (4-conv layer network and ResNet12) compared with a series of strong baselines.

**Strengths:**

- It is interesting to introduce prompt tuning to the MAML framework. Leveraging prompts to adapt input space accounting for the shared meta feature extractor is also orthogonal to the previous MAML-based methods.
- The proposed meta prompt is well-motivated and developed. While lacking strong empirical evidence (see weaknesses), the Bayesian treatment of prompt learning is provided and seems to work in practice.
- The experiment is designed well with a series of relevant baseline methods, a detailed ablation study, and necessary model discussions.

**Weaknesses:**

- While the motivation for introducing the Bayesian meta-prompt is clear and reasonable, the current experimental results cannot fully support it due to the lack of a large meta-feature extractor network (backbone). The main goal of this work is to leverage prompt tuning to avoid inefficient nested gradient calculation on large models; yet the backbones used in the experiment are too small to validate the effectiveness of the proposed method. *Table 2* also exacerbates the above concern, where all the methods actually report a comparable training/adaption time.
- The prompt poster is inconsistent through Eqs (1), (4), and (6). It remains confusing 1) where the meta prompt is sampled from and 2) what parameters are used in optimizing the prompt posterior. Is $p(s)$ in (1) the prompt prior or the posterior, or should it be written as $p(s|D)$? In meta-training, should the method sample prompt from the prior or the learned posterior? Comparing (4) and (6), is $q(s)$ optimized through $\phi$ or $\theta$?
- It lacks strong empirical evidence to indicate the effectiveness of the proposed Bayesian meta-prompt. From Table 3, the probabilistic formulation of the meta-prompt did not show a significant improvement. How about directly learning $s$ through a MAML approach?
- The lit review is insufficient. Some relevant works, e.g., Probabilistic Model-Agnostic Meta-Learning, should be clearly discussed in the paper.

**Questions:**

Please refer to the questions in the `Weaknesses`. Plus, the reviewer is curious about the following questions:
- Can the proposed method be applied to large models? Such as ResNet101, ViT-L/14, etc.
- How does the proposed method choose the prompt prior? Is the proposed method sensitive to prior choice?
- Can the paper show some visualizations of how the prompt uncertainty changes across different tasks?

---

> ### Author Response · Authors · 2023-11-18
> **Author Rebuttal by Authors (1/2)**
>
> We thank the reviewer for the detailed review. The followings are the responses to the reviewers’ concerns and questions:
>
> `Q1: the backbones used in the experiment are too small to validate the effectiveness of the proposed method;`
>
> A1: That is quite a good question. We would like to clarify that the reason why we do not post the result on the large-scale (e.g., vit) is that the inefficiency issue of the traditional methods (e.g., MAML) prevents their application on the large-scale architecture. Instead, we have conducted the experiments on vit-based-patch16-224 with the efficient meta-learning methods (i.e., ours and ANIL), and found that our proposed two methods achieve an increase of 3%/3.6% (52.07%/52.67%  v.s. 49.01%) with a similar computational consumption. As for the result of table2, the decreased gap (compared to table1) of the consumption between ours and the baselines is due to the decreased scale of the few-shot tasks, from 5-shot to 1-shot. We also conducted the 5-way 10-shot experiment and the experimental results demonstrate that our method achieves an increase of 1.1% and 1.6% while only requiring 40% and 23% computation consumption for training when compared to MAML. Another experiment under 20-way 1-shot also confirms the efficiency of our methods, where the result demonstrates that our method requires 62% consumption of MAML for training but with an increase of 4% on performance.
>
> `Q2: Some mistakes (e.g., \phi) in the equations`
>
> A2: We thank again for the reviewer’s detailed review, which really help to improve the quality of our paper. We would like to clarify that in our assumptions, we consider the distribution of s as a Gaussian distribution, as explicitly stated in A.1 of the original submission. Accordingly, the $q(s)$ is parameterised by $\phi = \{ \mu, \sigma \}$. To prevent any potential misunderstanding, we relocated the details pertaining to $q(s)$ from the appendix to the main text in the revised version.
>
> `Q3: The sampling of the meta-prompt.`
>
> A3: Before the explanation, we would like to clarify that the concepts of prior and posterior are relative and context-dependent, tied to specific datasets. In equation.1, we present the generative process for our entire framework, where the distribution of the meta-prompt, denoted as s, serves as a prior for unseen tasks. During the training stage, we leverage available training data to infer the posterior distribution of the meta-prompt across the training dataset. As mentioned in our original submission, the posterior distribution of the meta-prompt is intractable. Therefore, we introduce a variational distribution, denoted as $q(s;\phi)$, to approximate the posterior distribution $p(s|\mathcal{D})$. Subsequently, during the testing stage, we sample the meta-prompt from the posterior distribution (i.e., the variational distribution) to address few-shot tasks. The formulation of the Bayesian meta-prompt aligns with established approaches in the realm of Bayesian meta-learning, building upon existing works [1, 2].
>
> `Q4: It lacks strong empirical evidence to indicate the effectiveness of the proposed Bayesian meta-prompt. How about directly learning the meta-prompt through a MAML approach?`
>
> A4: The results obtained by directly learning the meta-prompts through a MAML approach are denoted as "w/o Bayes" in Table 3. Our experimental findings indicate that the non-Bayesian method experiences a decrease of nearly 1.5% in average performance across the four datasets. It is essential to note that the reported performance represents the average result across these datasets. For a more detailed breakdown of results on each dataset, please refer to Table. 8 and 9, which demonstrate that the Bayesian treatment consistently improves performance across all four datasets. Furthermore, the advantage of the Bayesian treatment extends beyond the experimental results. It encompasses the capability to handle noise and uncertainty when applying the method in real-world scenarios. This underscores the broader applicability and robustness of our Bayesian meta-prompt in addressing challenges that may arise in practical applications.

---

> ### Author Response · Authors · 2023-11-18
> **Author Rebuttal by Authors (2/2)**
>
> `Q5: Some lacked relevant papers;`
>
> A5: The paper [3] mentioned by the reviewer focuses on the problem of Bayesian meta-learning, which casts the task-specific parameters as the deterministic value and only operates the Bayesian treatment on the meta-parameters. We follow a similar idea but only use the Bayesian treatment on our proposed meta-prompt instead of all the meta-parameters. We will add this paper to the discussion about the Bayesian meta-learning in our paper.
>
> `Q6: Can the proposed method be applied to large models? Such as ResNet101, ViT-L/14, etc.`
>
> A6: It is definite that our method can be used on the large models. As we mentioned in the response to Q1, we have conducted the experiments on VIT-base-patch16-224, and found that our method achieves an improvement of 3%/3.6% (52.07%/52.67%  v.s. 49.01%) with a similar computational consumption, when compared to ANIL. Unfortunately, according to the inefficiency issue, it is difficult to train a vit-based model with MAML in an acceptable time.
>
> `Q7: How does the proposed method choose the prompt prior? Is the proposed method sensitive to prior choice?`
>
> A7: To answer this question, we conduct the experiments with different priors, i.e., the mixture Gaussian distribution used in our paper, and the standard normal distribution (the mean is 0 and the standard deviation is 1). The experimental results show that the meta-prompt with a standard normal distribution achieves a comparable performance under 5-shot setting while seeing a decrease of 0.7%-1.8% under 1-shot, compared to the meta-prompt with a mixture prior(our reported method in the original submission). It is reasonable that the mixture prior helps to focus more on the long-tail data, especially under 1-shot setting where it is easily affected by some outliers, according to the existing work [4].
>
> `Q8: Can the paper show some visualizations of how the prompt uncertainty changes across different tasks?`
>
> A8: We guess the reviewer would like to see the task-specific prompts across different tasks. We have visualised the task-specific prompts across different 5-way 5-shot tasks in Figure 6 of the appendix.
>
> [1] Meta-Learning Probabilistic Inference For Prediction. ICLR 2019.
>
> [2] Variational Continual Bayesian Meta-Learning. NeurIPS 2021.
>
> [3] Probabilistic Model-Agnostic Meta-Learning. NeurIPS 2018.
>
> [4] Weight uncertainty in neural network. ICML 2015.

---

### Official Review · Reviewer_WuMv · 2023-11-01

**Soundness:** 3 good
**Presentation:** 3 good
**Contribution:** 2 fair
**Rating:** 5
**Confidence:** 3

**Summary:**

This paper proposes a novel efficient gradient-based meta-learning framework that freezes the model backbone and only updates task-specific "prompts" to extract meta-knowledge for few-shot tasks. The task-specific "prompt" is constructed through the meta learning bi-level idea again based on a learnable Bayesian meta-"prompt". Experiments are conducted to validate the efficiency and effectiveness of the proposed approach.

**Strengths:**

1. The motivation is clear.
2. The presentation is good and not hard to follow.
3. Extensive experiments demonstrate the efficiency and effectiveness of the proposed approach.

**Weaknesses:**

1. The technical novelty might not be enough. Although the paper claims that the idea is inspired from prompt tuning of LLM, it is still to meta-learn part of model parameters (the majority of model parameters are freezed), so that the training becomes efficient. Similar papers are:
[1] Rapid learning or feature reuse? towards understanding the effectiveness of maml, ICLR 2019
[2] Boil: Towards representation change for few-shot learning, ICLR 2020

2. minor issues: one ")" is missed in Eq.10.

**Questions:**

I am still not convinced by the connection between prompt tuning and the idea in the paper. The idea of this paper is more close and similar to ANIL and BOIL [1,2], and I did not find a similarity with prompt tuning, technically speaking. Since the authors use many spaces to explain prompt tuning, I assume I missed their connection. If the authors could explain this connection and key difference from ANIL and BOIL in the response, I am open to raising the rating.
[1] Rapid learning or feature reuse? towards understanding the effectiveness of maml, ICLR 2019
[2] Boil: Towards representation change for few-shot learning, ICLR 2020

---

> ### Author Response · Authors · 2023-11-18
> **Author Rebuttal by Authors (1/1)**
>
> We thank the reviewer for the detailed review and the insightful questions. The followings are the responses:
>
> `Q1: The connection between prompt tuning and our work`
>
> A1: Prompt tuning has recently emerged as a parameter-efficient fine-tuning technique within the pre-training followed by fine-tuning paradigm. Its fundamental principle involves knowledge extraction from a pre-trained model, as opposed to the conventional approach of adapting this knowledge for downstream tasks through full fine-tuning. On the other hand, meta-learning entails acquiring meta-knowledge from existing training tasks, thereby facilitating quick adaptation to few-shot tasks. The prevailing gradient-based meta-learning method, designed to acquire an optimal initialisation, formulates meta-knowledge as neural network weights, akin to the pretraining-finetuning paradigm. Building upon the insights from prompt tuning, our research endeavors to employ task-specific prompts for efficient guidance of a fixed backbone (shared meta-knowledge) in handling few-shot tasks. Notably, it stands in contrast to the resource-intensive paradigm of adapting meta-knowledge that is akin to the fine-tuning process applied to the entire model.
>
> `Q2: The key difference between our work and ANIL/BOIL`
>
> A2: One of our contributions is to change the costly meta-knowledge adaption paradigm in traditional meta-learning into efficient meta-knowledge extraction. As we claimed in the introduction, our work employs the fixed backbone, which is supported by the previous ANIL that confirms the effect of the shared features (feature reuse in the original paper). However, ANIL faces limitations, particularly in its inability to address the complexities of certain few-shot tasks, as highlighted by BOIL. For example, the shared feature extractor cannot deal with heterogeneous scenarios where different tasks might require different features. Instead, BOIL proposes adapting the feature extractor for specific tasks with a fixed classification head, yet it incurs a computational cost similar to that of MAML, as evidenced in our experiments. Differently, our work seeks to employ a small number of parameters (task-specific prompts in our work) to prompt the fixed feature to deal with different few-shot tasks in a more efficient way. **The experimental results confirm the superior performance of our method compared to BIOL, while requiring a similar consumption as ANIL.**
>
> `Q3: The technical novelty might not be enough.`
>
> A3: In addition to the meta-knowledge extraction paradigm mentioned above, we also provide two other contributions. Firstly, we theoretically analyse the role of the learned uncertainty of our learnable Bayesian meta-prompt. This exploration contributes a foundational understanding of the mechanisms for constructing the task-specific prompts in our methodology. Secondly, we propose two modulation methods to construct the task-specific prompt from the meta-prompt. These methods utilise the learned posterior distribution of the meta-prompt to modulate the knowledge transfer among tasks. The aim of our work is to address the efficiency challenges that have impeded the application of meta-learning in real-world scenarios. By combining these contributions, we aspire to provide valuable insights to enhance the applicability of meta-learning in practical settings.
>
> Moreover, the mentioned minor typos are revised in the updated version.

---

### Official Review · Reviewer_DzXH · 2023-11-06

**Soundness:** 3 good
**Presentation:** 2 fair
**Contribution:** 3 good
**Rating:** 5
**Confidence:** 2

**Summary:**

The manuscript proposes a meta-learning method that is based on meta-knowledge extraction, where prompt learning is used to link the general knowledge and task-specific knowledge.  The task-specific prompts are used to extract meta-knowledge for few-shot tasks, and a Bayesian meta-prompt provides a better initialization for them (like the task-specific parameter and the initialization of the model parameter in model agnostic meta-learning (MAML)). The manuscript proposes two gradient-based update rules to update the task-specific prompts from the meta-prompts using the uncertainty captured by the standard deviations of the posteriors of the meta-prompts (hence why the proposed method is called uncertainty-aware prompted meta-learning). The proposed methods are compared with multiple meta-learning methods on several few-shot learning datasets.

**Strengths:**

Overall, the manuscript is well written and explains the motivation for using of the prompt learning with sufficient background.

As a prompt learning method (the knowledge extraction), the proposed methods may provide a way to improve the computation load that meta-knowledge adaptation methods would have.

**Weaknesses:**

The main proposed methods (for constructing the task-specific prompts from the meta-prompt) look ad hoc, although the authors claim to provide the theoretical analysis supporting the methods. I understood that the proposed two gradient-based update rules (in eq. (11)) are not designed to directly optimize the something derived from a full Bayesian model including the meta-prompts and task-specific prompts (like eq. (6)). The fact that the two update rules takes different inputs from the posteriors of the meta-prompts (the hard modulation takes the means and the standard deviations, but the soft modulation takes only the standard deviations) makes both update rules look more ad hoc.

I think the manuscript could be improved in presentation. There are missing details and typos in the main text. For example:
1)  The exact definition of the Gaussian distributions for the meta-prompts is not given in the main text. (as well as the definition of the standard deviation of d-th dimension of the meta-prompt, \sigma_d). I could find some descriptions in the appendix, but I think these descriptions should be included in the main text for easier reading.
2) In Section 5.2 (RQ2), the figure and table are referenced incorrectly (Figure 1 and Table 1?); what is the exact definition of the (weighted) KL-term considered in this section? Could it be eq. (4)? What is the exact meaning of the removal of the Bayesian treatment?

Minor comments:
Figure 1 is not directly mentioned in the main text?
The symbol \mathcal{L} was introduced as a loss function but also was used as the lower bound (which should be maximized) in eq. (6). Please also check eq. 5 (the relationship between the loss function and the likelihood).

**Questions:**

The manuscript states that one of the main motivations (of using the prompt learning) is to improve the computational inefficiency of the meta-knowledge adaptation. However, in the experimental result section, the improvements in the computation time do not look significant compared to the baseline (e.g., MAML) for the choice of the backbone network (e.g., Conv4). This could lead to the misunderstanding that the improvements reported in the experimental results are a matter of implementation (e.g., code optimization). The datasets may not be large enough to contrast the improvements in the computation time of the methods?

In Section 5.2 (RQ2), what did you intend to show the performance changes in the dimension of the meta prompts? It does not clear from the text. How can we understand this pattern in terms of Bayesian learning? In the current version of the manuscript, the figures just say that the dimension of the meta prompts is also the hyperparameter to be tuned by the users.

---

> ### Author Response · Authors · 2023-11-18
> **Author Rebuttal by Authors (1/2)**
>
> We thank the reviewer for the comprehensive review and the insightful questions. The followings are the responses:
>
> `Q1: The proposed methods (for constructing task-specific prompts) look ad hoc.`
>
> A1:  We would like to clarify that the difference in using the mean or not for constructing task-specific prompts depends on the modulation mechanism. Specifically, the soft way employs the input directly to modulate the learning rate, while the hard way utilizes the input to decide whether to update a certain dimension or not. Consequently, the soft way is more sensitive to the value of the input. easily influenced by some outliers. The unlimited scale of the mean value limits its application on the direct modulation in the soft way, although the mean value indicates the role of the dimension where the larger means more important or effective the dimension is to the prediction given a specific task (claimed in our original submission according to [1]).
>
> We also conducted the experiment (see the following table) and found that the input of the mean leads to a decrease of 0.5%-2% (especially for the 1-shot setting), which also confirms the negative use of the mean value in the soft way. We will clarify this point in the revised paper.
>
> |    |Conv4|    |Resnet12||
> |-|-|-|-|-|
> |    |1-shot|5-shot|1-shot|5-shot|
> |Reference| 56.84$\pm$0.73| 71.96$\pm$0.59 | 58.52$\pm$0.77| 73.77$\pm$0.57|
> |Soft Modulation with Mean|56.35$\pm$0.73 | 71.26$\pm$0.81 | 56.70$\pm$0.77| 73.07$\pm$0.81|
>
> `Q2: Some missing details and typos in the paper.`
>
> A2: Thank you again for pointing out the typos in our original version. We moved the definition of the variational distribution into the main text. In addition, the incorrect references in RQ2, the missing reference for Fig.1 and the incorrect definition in eq.5 and 6 mentioned in the minor comments are fixed in the updated version.
>
> `Q3: what is the exact definition of the (weighted) KL-term considered in this section? Could it be eq. (4)? What is the exact meaning of the removal of the Bayesian treatment?`
>
> A3: The KL-term explicitly corresponds to the latter terms in Equation (4), and we have incorporated the relevant reference in the updated version. The meaning of the removal of the Bayesian treatment means that we model the meta-prompt in a deterministic way instead of a probabilistic way. In other words, we treat the meta-prompt as a deterministic vector instead of a probabilistic distribution. This Bayesian treatment enables to provide a flexible initialisation for constructing task-specific prompts.
>
> `Q4: The improvements in the computation time do not look significant compared to the baseline (e.g., MAML) for the choice of the backbone network (e.g., Conv4).`
>
> A4: The efficiency of our methods mainly results from the fixed backbone when dealing with specific tasks, as analysed in the appendix A.3 of the original submission. Since Conv-4 is just a simple backbone, the computational consumption incurred by the backbone remains relatively modest but our method still achieves a 58% decrease compared to MAML (refer to Table 1). In the case of more complex backbones, such as Res12, our approach demands only 24% of the computational resources required by MAML. Moreover, we have also conducted the experiments on a larger backbone (i.e., vit_base_patch16_224), and our proposed two methods achieve an increase of 3%/3.6% (52.07%/52.67%  v.s. 49.01%) on the performance with a similar computational consumption, when compared to ANIL, an efficient baseline. As for MAML, the meta-knowledge adaption, we found it difficult to train such a large architecture in an acceptable time.

---

> ### Author Response · Authors · 2023-11-18
> **Author Rebuttal by Authors (2/2)**
>
> `Q5: The datasets may not be large enough to contrast the improvements in the computation time of the methods?`
>
> A5: The heterogeneous dataset used in our experiments contains four datasets (i.e., *Aircraft, CIFAR-FS, Mini-Imagenet and MiniQuickDraw*) with 40k images, which in our opinion should be adequately large in the context of few-shot learning. We guess the reviewer may be concerned about the scale of the task (e.g., 5-shot and 1-shot). In response, we have expanded our experimentation to include 5-way 10-shot scenarios. Notably, our method exhibits performance improvements of 1.1% and 1.6% while simultaneously achieving a significant reduction in training computational consumption (only requires 40% and 23% of MAML) for the Conv4 and Resnet12 backbones, respectively. Moreover, we also conducted another experiment with a 20-way 1-shot setting, to further confirm the efficiency of our proposed methods. Compared to the 5-way 1-shot setting, the computational consumption of our method decreases from 71% MAML (in Table 2) to 62%, with the increase of the number of the way, where our proposed two methods still see an increase of 4%/7.5% (31.04%/34.50% v.s. 27.05%) compared to MAML.
>
> `Q6: what did you intend to show the performance changes in the dimension of the meta prompts?`
>
> A6: In the RQ2, our objective is to investigate the impact of meta-prompts on performance. One notable attribute of the meta-prompt under consideration is its number of dimensions, a factor intricately linked to computational consumption, particularly in terms of the Bayesian treatment. Our experimental findings consistently reveal an anti-intuitive trend: larger prompts do not necessarily yield superior performance. Such an anti-intuitive conclusion offers an insight that finding an appropriate number of dimensions is more important than just simply adding the dimension.
>
> [1] Weight uncertainty in neural network. ICML 2015.

---

### Meta-Review · Area_Chair_Rf1n · 2023-12-07

**Metareview:**

The paper proposes an Uncertainty-Aware Prompt Meta-learning (UAPML) framework, which improves the efficiency of gradient-based meta-learning by focusing on meta-knowledge extraction for few-shot tasks instead of computationally expensive meta-knowledge adaption. The soft and hard modulation techniques are developed to automatically generate task-specific prompts while considering the shared and task specific information among the few-shot tasks.  Experiments are conducted in various settings to demonstrate the effectiveness of the proposed UAPML framework.

As pointed out by the reviewers, some key pieces of the proposed methodology are not clearly described, which lead to confusions around several important technical components. The evaluation results are not totally convincing as the improvements in the computation time are less significant compared to the baselines and experiments on large backbones are also missing. The paper should also more clearly highlight the distinctions from important related works (as pointed by the reviewers) to better justify the technical novelty.

**Justification For Why Not Higher Score:**

The evaluation results are not entirely convincing and there are confusions about some key technical components, making it difficult to assess the overall contribution and novelty.

**Justification For Why Not Lower Score:**

N/A

---

### Decision · Program_Chairs · 2024-01-16

Reject